# Learning Domain-Aware Task Prompt Representations for Multi-Domain All-in-One Image Restoration

**Guanglu Dong**[1,*]  **Chunlei Li**[2]  **Chao Ren**[1]  **Jingliang Hu**[2]  **Yilei Shi**[2]

**Xiao Xiang Zhu**[3]  **Lichao Mou**[2,†]

[1]Sichuan University  [2]MedAI Technology (Wuxi) Co. Ltd.  [3]Technical University of Munich

dongguanglu@stu.scu.edu.cn  lichao.mou@medimagingai.com

## Abstract

Recently, significant breakthroughs have been made in all-in-one image restoration (AiOIR), which can handle multiple restoration tasks with a single model. However, existing methods typically focus on a specific image domain, such as natural scene, medical imaging, or remote sensing. In this work, we aim to extend AiOIR to multiple domains and propose the first multi-domain all-in-one image restoration method, DATPRL-IR, based on our proposed *Domain-Aware Task Prompt Representation Learning*. Specifically, we first construct a task prompt pool containing multiple task prompts, in which task-related knowledge is implicitly encoded. For each input image, the model adaptively selects the most relevant task prompts and composes them into an instance-level task representation via a prompt composition mechanism (PCM). Furthermore, to endow the model with domain awareness, we introduce another domain prompt pool and distill domain priors from multimodal large language models into the domain prompts. PCM is utilized to combine the adaptively selected domain prompts into a domain representation for each input image. Finally, the two representations are fused to form a domain-aware task prompt representation which can make full use of both specific and shared knowledge across tasks and domains to guide the subsequent restoration process. Extensive experiments demonstrate that our DATPRL-IR significantly outperforms existing SOTA image restoration methods, while exhibiting strong generalization capabilities. Code is available at https://github.com/GuangluDong0728/DATPRL-IR.

## 1 Introduction

Image restoration (Guo et al., 2024; Chen et al., 2022; Zamir et al., 2022; Liang et al., 2021; Zamir et al., 2021) has long been a fundamental research in computer vision, aiming to recover high-quality images from their degraded versions. With the advancement of deep learning, image restoration has found widespread applications across multiple image domains, including natural scene, medical imaging, and remote sensing. Early explorations mainly focused on designing independent models for different tasks within each domain, such as natural image super-resolution (SR) (Dong et al., 2014), natural image deraining (Dong et al., 2025b), natural image deblurring (Chakrabarti, 2016), CT denoising (Chen et al., 2017), MRI SR (Chen et al., 2018), PET synthesis (Luo et al., 2022), remote sensing image (RSI) SR (Lei et al., 2017), RSI cloud removal (Liu et al., 2025), RSI dehazing (Shen et al., 2020), etc.. However, training separate models is undoubtedly time-consuming and resource-intensive, and this greatly limits their applicability in complex real-world scenarios.

To address the above challenge, all-in-one image restoration (AiOIR) (Li et al., 2022; Cui et al., 2024; Potlapalli et al., 2023; Zamfir et al., 2025; Conde et al., 2024; Zhang et al., 2025) has gained increasing attention in recent years, as it seeks to provide a unified solution for handling multiple

---

*Work done during an internship at MedAI Technology (Wuxi) Co. Ltd.

†Corresponding author.

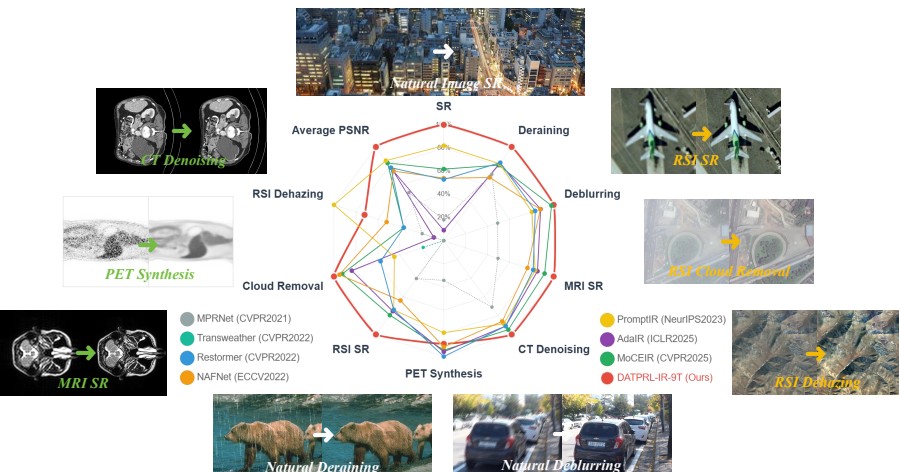

Figure 1: This paper makes a preliminary exploration of multi-domain all-in-one image restoration (MD-AiOIR), aiming at further extending the restoration capability of a single model to a broader range of tasks and image domains, including natural scene, medical imaging, and remote sensing.

restoration tasks with a single model. AiOIR first emerge in natural scene, by leveraging implicit or explicit prompts (Potlapalli et al., 2023; Conde et al., 2024), contrastive learning (Li et al., 2022), degradation classification (Hu et al., 2025), prior information (He et al., 2024), or mixture-of-experts (MoE) architecture (Zamfir et al., 2025) to enable the restoration networks to better distinguish between different tasks. Meanwhile, inspired by the progress in natural scene, AiOIR methods have also gained popularity in medical imaging (Chen et al., 2025; Yang et al., 2025b; 2024a). Though existing methods have achieved remarkable success, they primarily focus on maximizing task differences within a single domain and often overlook the shared commonalities across tasks. They also do not consider the differences or connections between different image domains. When faced with more restoration tasks and image domains, they will face increased learning difficulty.

In this work, we make the first exploration of multi-domain all-in-one image restoration (MD-AiOIR), aiming to unify diverse restoration tasks across multiple domains within a single model. Inspired by the concept of prompt pool in L2P (Wang et al., 2022c), we propose domain-aware task prompt representation learning (DATPRL), which adopts a dual-prompt-pool design to learn prompt representations that carry both task-relevant and domain-relevant knowledge. Based on DATPRL, we introduce the first MD-AiOIR method, DATPRL-IR. Specifically, we first construct a task prompt pool with numerous task prompts. For each input image, our DATPRL-IR can adaptively select the most relevant task prompts through a similarity based query mechanism. To express more diverse instance-level information, we propose a prompt composition mechanism (PCM) to combine the selected task prompts into a task prompt representation. The task prompts are optimized jointly with the restoration objectives, ensuring the learning of task-specific knowledge while allowing knowledge sharing across tasks. Additionally, to endow the model with domain awareness, we build a separate domain prompt pool to store domain-related knowledge. We leverage the powerful image understanding ability of multimodal large language models (MLLMs) and employ a cross-modal alignment to distill domain priors from MLLMs (e.g., LLaVA (Liu et al., 2024)) into the domain prompts. Similarly, our DATPRL-IR will adaptively select the most relevant domain prompts for each input image, and then apply PCM to combine them into an instance-level domain prompt representation. The two prompt representations are then fused into the final domain-aware task representation to guide the subsequent restoration process. Our method effectively exploits the shared knowledge across different tasks and domains, significantly reducing the learning difficulty and facilitating performance improvement across tasks. As illustrated in Figure 1, under the guidance of domain-aware task prompt representations, our DATPRL-IR significantly surpasses existing methods, demonstrating strong generalization capability.

Our main contributions can be summarized as follows: (1) To the best of our knowledge, we propose the first multi-domain all-in-one image restoration method, DATPRL-IR, which can handle diverse restoration tasks across multiple domains. (2) Through the proposed domain-aware task prompt

representation learning, our DATPRL-IR effectively leverages both specific and shared knowledge across tasks and domains to guide the restoration process. (3) Extensive experiments demonstrate that our method outperforms existing SOTA image restoration approaches on MD-AiOIR tasks.

## 2 RELATED WORK

**Single-Task Image Restoration.** With the development of deep learning (LeCun et al., 2015), image restoration techniques have made continuous progress across multiple imaging domains, including natural scene (Dong et al., 2014; Zhang et al., 2017; Dong et al., 2025b; Li et al., 2025b), medical imaging (Chen et al., 2018; 2017; Luo et al., 2022), and remote sensing (Lei et al., 2017; Wu et al., 2023; Liu et al., 2025). By leveraging specific designs for different domains and tasks, a wide variety of restoration sub-tasks have flourished. Recently, with the growing demand for multi-task image restoration and the continuous evolution of foundation backbones (e.g., CNNs (He et al., 2016; Ronneberger et al., 2015), Transformers (Vaswani et al., 2017; Dosovitskiy et al., 2020), and Mamba (Gu & Dao, 2023; Zhu et al., 2024)), a series of general image restoration baselines (Zamir et al., 2021; Liang et al., 2021; Zamir et al., 2022; Wang et al., 2022a; Chen et al., 2022; Guo et al., 2024; Li et al., 2025a) have also emerged, which are capable of handling diverse types of degradations within a unified model architecture. However, these methods require training a separate model for each individual task, which is time-consuming and labor-intensive.

**All-in-One Image Restoration.** To overcome the limitations above, various all-in-one image restoration (AiOIR) frameworks (Li et al., 2022; Cui et al., 2024; Conde et al., 2024; Potlapalli et al., 2023; Zamfir et al., 2025) have continuously emerged and achieved sustained breakthroughs, especially in the natural image domain. AirNet (Li et al., 2022) is the first to achieve AiOIR through contrastive learning (Chen et al., 2020; He et al., 2020). IDR (Zhang et al., 2023) integrates degradation-specific priors into the restoration process to enhance performance. PromptIR (Potlapalli et al., 2023) uses learnable prompt components to encode different degradation information. DA-CLIP (Luo et al., 2023) decouples degradation and content semantics based on CLIP (Radford et al., 2021), making the model more sensitive to various degradation knowledge. InstructIR (Conde et al., 2024) guides the image restoration model through human-written instructions. MoCE-IR (Zamfir et al., 2025) introduces complexity experts within a mixture-of-experts (MoE) architecture to efficiently allocate task-specific resources. DCPT (Hu et al., 2025) propose a degradation classification pre-training strategy to classify the degradation type of input images. In addition to the methods for natural domain, recent AiOIR techniques have also started to gain attention in the field of medical imaging (Yang et al., 2024a; 2025b; Chen et al., 2025). However, current research mainly focuses on exploring a specific domain, with most approaches aiming to better distinguish between different tasks while overlooking the commonalities between them.

**Prompt Learning-based Image Restoration.** Inspired by the success of prompt learning in natural language processing (Shin et al., 2020; Brown et al., 2020), high-level computer vision (Wang et al., 2022c;b), and multi-modal models (Zhou et al., 2022; Yao et al., 2023), it has also been widely applied in image restoration recently. PromptRestorer (Wang et al., 2023a) takes the advantage of prompt learning to perceive degradation, achieving progress on individual tasks such as image deraining, deblurring, and dehazing. SFD (Dong et al., 2025a) trains learnable antonymous prompt pairs in an adversarial manner to promote global discrimination for super-resolution images. PromptIR (Potlapalli et al., 2023) is the first to explore the capability of prompt learning in all-in-one image restoration, and it subsequently inspires a series of prompt-based all-in-one restoration methods (Gao et al., 2024; Kong et al., 2024; Wu et al., 2025; Ma et al., 2023; Ai et al., 2024; Conde et al., 2024), which employ explicit or implicit prompts to guide restoration process. In this work, different from existing methods, we propose a novel domain-aware task prompt representation learning method, which effectively leverages both both the specific and shared knowledge across restoration tasks and image domains to guide multi-domain all-in-one image restoration.

## 3 DATPRL-IR FOR MD-AiOIR

### 3.1 DOMAIN-AWARE TASK PROMPT REPRESENTATION LEARNING

**Motivation.** In this work, we aim to take the first step towards Multi-Domain All-in-One Image Restoration (MD-AiOIR), extending AiOIR to more restoration tasks across multiple image do-

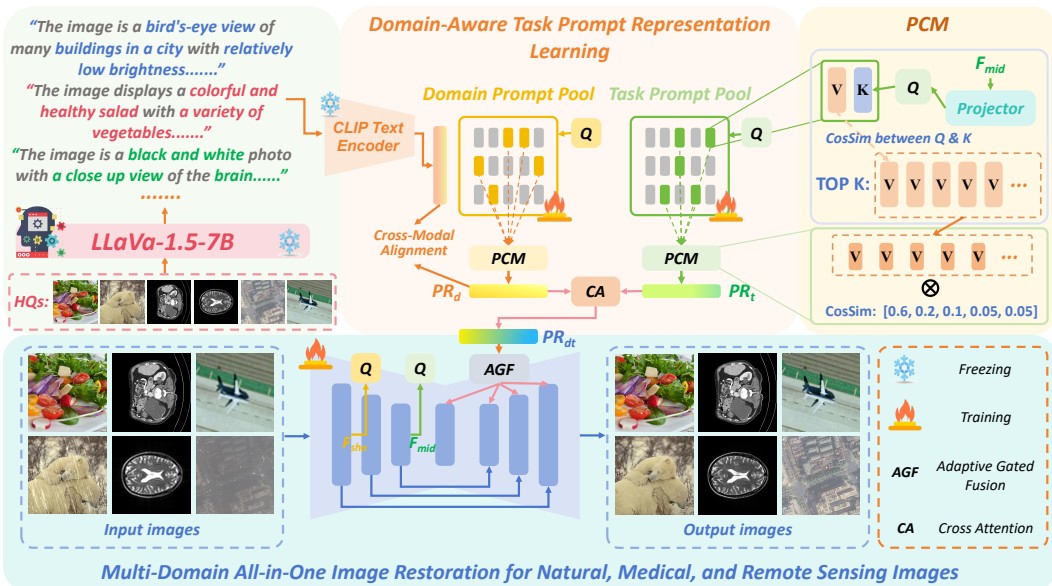

Figure 2: Framework of the proposed DATPRL-IR for multi-domain all-in-one image restoration. By introducing domain-aware task representation learning, DATPRL-IR can fully utilize both specific and shared knowledge across tasks and domains, effectively reducing the learning difficulty of the model and improving its performance.

mains. A key challenge is how to alleviate the learning difficulties introduced by the increasing number of restoration tasks and image domains. Prior studies (Chen et al., 2024; Zhang et al., 2023) have found that different image restoration tasks share certain inherent commonalities or similar latent representations and there is a certain mutual promotion effect between different restoration tasks (Conde et al., 2024), such as super-resolution and motion deblurring. Additionally, though images from different domains exhibit their own distinct visual characteristics, they share some common terms. Combining these specific and shared visual characteristics can facilitate the discrimination of the image domain. For instance, "grayscale + human organs" typically corresponds to medical images, while "bird's-eye view + buildings" often indicates remote sensing scenarios. Therefore, we infer that effectively leveraging both the specific and shared knowledge across tasks and domains can help reduce the learning difficulty and further enhance the restoration performance. Inspired by L2P (Wang et al., 2022c), prompt pools offer an effective way to encode and organize both specific and shared knowledge. Building on this insight, we propose *Domain-Aware Task Prompt Representation Learning* and introduce the first MD-AiOIR method, DATPRL-IR.

**Overall Framework.** As illustrated in Figure 2, our DATPRL-IR mainly consists of an encoder–decoder based restoration backbone, a task prompt pool, a domain prompt pool, a CLIP (Radford et al., 2021) text encoder, and the LLaVA-1.5-7B (Liu et al., 2024) model. The task and domain prompt pools store $N_t$ and $N_d$ prompts, respectively. These prompts implicitly encapsulate knowledge related to restoration tasks and image domains. Given a degraded input image, our model adopts a query–retrieval–composition paradigm to adaptively query both prompt pools to retrieve the most relevant prompts for the task and domain, which are then composed into two representations: a task prompt representation $\mathbf{PR}_t$ and a domain prompt representation $\mathbf{PR}_d$. Subsequently, these two representations are integrated through a cross-attention mechanism to produce a domain-aware task prompt representation $\mathbf{PR}_{dt}$, which can effectively guide the restoration process.

**Task Prompt Pool.** Task prompt (TP) pool is used to implicitly store both specific and shared knowledge across different tasks, and each prompt in TP pool is represented as a pair of a key $\mathbf{K}_j^{\text{task}} \in \mathbb{R}^d$ and value $\mathbf{V}_j^{\text{task}} \in \mathbb{R}^{T \times d}$. We use a learnable projector to map the middle feature $F_{mid}$ of the input image extracted by the encoder of the restoration network into a query $\mathbf{Q}^{\text{task}} \in \mathbb{R}^d$ with the same dimension as $\mathbf{K}_j^{\text{task}}$. Based on the cosine similarities $s_j^{\text{task}}$ between the $\mathbf{Q}^{\text{task}}$ and each $\mathbf{K}_j^{\text{task}}$, the top k most relevant values $\mathbf{V}_k^{\text{task}}$ can be retrieved from the TP pool. To enable the limited set of prompts to provide more diverse instance-level prompt guidance, we design a prompt composition

| Word Cloud of DF2K | Word Cloud of GoPro | Word Cloud of CUHK | Word Cloud of AAPM | Word Cloud of IXI MRI | T-SNE of text features |

Figure 3: A partial visualization of the word clouds generated from the text descriptions produced by LLAVA, and the t-SNE clustering analysis of the text descriptions corresponding to the 9 datasets from different domains and tasks. It can be observed that images from different domains exhibit their own characteristics while also sharing certain overlapping features.

mechanism (PCM) to combine the selected $\mathbf{V}_k^{\text{task}}$ into an instance-level task prompt representation $\mathbf{PR}_t$ according to the similarity scores $s_k^{\text{task}}$:

$$\alpha_j^{\text{task}} = \frac{\exp(s_j^{\text{task}}/T_{\text{task}})}{\sum_{\ell \in k} \exp(s_\ell^{\text{task}}/T_{\text{task}})}, \quad \mathbf{PR}_t = \sum_{j \in k} \alpha_j^{\text{task}} \mathbf{V}_j^{\text{task}}. \tag{1}$$

where $\alpha_j^{\text{task}}$ denotes the relative cosine similarity among the selected prompts, and $T_{\text{task}}$ is the temperature parameter. During training, task prompts are optimized jointly with restoration objectives, ensuring the learning of task-related knowledge while allowing knowledge sharing across tasks.

**Domain Prompt Pool.** As shown in Figure 3, images from different domains exhibit their own characteristics while also sharing certain overlapping features. Integrating these characteristics will help the model to better determine the domain of an input image and to enrich its domain knowledge. To this end, we construct another domain prompt (DP) pool to store and organize such domain priors. The domain prompts in the DP Pool are also constructed as a set of key–value pairs. For each input image, we use another projector to map the shallow features extracted from the first layer of the restoration backbone into a domain query $\mathbf{Q}^{\text{dom}}$. Based on the cosine similarity between $\mathbf{Q}^{\text{dom}}$ and the key $\mathbf{K}_j^{\text{dom}}$ of each domain prompt, we select the top k most relevant prompts. Similar to the TP pool, we also employ the PCM to combine the value $\mathbf{V}_j^{\text{dom}}$ of these candidates into an instance-level domain prompt representation $\mathbf{PR}_d$. To endow $\mathbf{PR}_d$ with rich and interpretable domain knowledge, we employ LLaVA-1.5-7B to generate multi-perspective descriptions for the high-quality images (HQs) corresponding to each input, covering aspects such as image content, color richness, object category, brightness, and camera/viewpoint. During training, these textual descriptions are fed into the CLIP text encoder to obtain the corresponding text features $\mathbf{F}_{\text{text}}$, and the training of the DP Pool is constrained by the following cross-modal alignment loss:

$$\mathcal{L}_{\text{align}} = \frac{1}{B} \sum_{n=1}^{B} \left( 1 - \cos(\mathbf{PR}_{\text{dom}}^n, \mathbf{F}_{\text{text}}^n) \right). \tag{2}$$

where $B$ denotes the batch size. During the joint training of our model, $\mathcal{L}_{\text{align}}$ encourages the domain prompts to capture both specific and share domain knowledge that benefits the restoration objective, providing domain-related information to the network. It is worth noting that the LLaVA and CLIP will not be used during the inference stage without introducing any additional inference overhead.

**Domain-Aware Task Prompt Representation.** Finally, the task and domain prompt representations $\mathbf{PR}_t$ and $\mathbf{PR}_d$ will be fused through a cross-attention layer to learn a domain-aware task prompt representation $\mathbf{PR}^{\text{dt}}$, which will be used to guide the restoration process. Considering that different layers of the restoration backbone may have varying demands for prompt information, inspired by UniECS (Liang et al., 2025), we dynamically control the contribution ratio between backbone features and $\mathbf{PR}_{\text{dt}}$ at each layer through an adaptive gated fusion (AGF):

$$\mathbf{F}_l^e = \text{CrossAttn}(\alpha_l \mathbf{F}_l, \ (1 - \alpha_l) \mathbf{PR}_{\text{dt}}), \tag{3}$$

where $\mathbf{F}_l^e$ and $\mathbf{F}_l$ denote the enhanced feature map and the pre-fusion feature map at the $l$-th layer respectively, and $\alpha_l \in [0, 1]$ is a learnable gating coefficient for the $l$-th layer. AGF allows each layer to independently learn the optimal fusion ratio, enabling a adaptive integration of $\mathbf{PR}^{\text{dt}}$ and $\mathbf{F}_l$.

## 3.2 PROMPT POOL REGULARIZATION

We introduce a series of regularization terms to avoid the model degenerating into undesirable behaviors: e.g., over-relying on a small subset of prompts, or learning redundant or highly correlated prompt contents, especially during the early training phase. Firstly, we adopt a diversity regularization to encourage diversity of the learned prompts. Given a prompt pool with $N$ prompts, we first compute the pairwise cosine similarity matrix of their values $\mathbf{V} \in \mathbb{R}^{T \times d}$:

$$\mathbf{S}_{ij} = \frac{\mathbf{V}_i \cdot \mathbf{V}_j^{\mathbf{T}}}{\|\mathbf{V}_i\|_2 \|\mathbf{V}_j\|_2}, \tag{4}$$

where $\mathbf{S}_{ij}$ is the pairwise cosine similarity. To exclude self-similarity, we apply a mask $\mathbf{M} = \mathbf{I} - \mathbf{I}_N$, where $\mathbf{I}_N$ is the $N \times N$ identity matrix. The diversity regularization loss is then formulated as:

$$\mathcal{L}_{\mathrm{div}} = \frac{1}{N(N-1)} \sum_{i=1}^{N} \sum_{j=1}^{N} \mathbf{M}_{ij} \cdot \max\left(0, \mathbf{S}_{ij} - \tau_{\mathrm{div}}\right), \tag{5}$$

where $\tau_{\mathrm{div}}$ is a predefined similarity threshold. Minimizing $\mathcal{L}_{\mathrm{div}}$ encourages prompts to occupy distinct regions in the representation space, avoiding collapse to similar contents.

Furthermore, we adopt a prompt entropy regularization to encourage more balanced utilization across the prompts. Given a query $\mathbf{q} \in \mathbb{R}^d$ mapped from the input image and a pool of $P$ prompt keys $\{\mathbf{k}_j\}_{j=1}^{P}$, we first compute the selection probability $p_j$ of each prompt based on the cosine similarity score $s_j$ between $\mathbf{q}$ and $\mathbf{k}_j$. The selection probabilities of each prompt are obtained via a softmax and their entropy is computed as:

$$p_j = \frac{\exp(s_j)}{\sum_{m=1}^{P} \exp(s_m)}, \quad H(\mathbf{p}) = -\sum_{j=1}^{P} p_j \log p_j, \tag{6}$$

where $H(\mathbf{p})$ denotes the entropy of the probability distribution. Then the balance loss is defined as:

$$\mathcal{L}_{\mathrm{bal}} = \log P - H(\mathbf{p}). \tag{7}$$

$\mathcal{L}_{\mathrm{bal}}$ encourages balanced prompt utilization during training. In addition, to enhance the sensitivity of instance-level prompt selection, we apply a contrastive regularization $\mathcal{L}_{con}$ as detailed in the Appendix C. All regularization terms are applied to both the two prompt pools.

## 3.3 OVERALL OPTIMIZATION OBJECTIVE

The final training objective combines the primary reconstruction loss with cross-modal alignment loss and prompt regularization terms, and the total loss can be formulated as follows:

$$\mathcal{L} = \underbrace{\lambda_{pix}\mathcal{L}_{pix} + \lambda_{fft}\mathcal{L}_{fft}}_{\text{Reconstruction Loss}} + \underbrace{\lambda_{align}\mathcal{L}_{align} + \lambda_{div}\mathcal{L}_{div} + \lambda_{bal}\mathcal{L}_{bal} + \lambda_{con}\mathcal{L}_{con}}_{\text{Cross-Modal Alignment and Prompt Regularization}}, \tag{8}$$

where $\mathcal{L}_{pix}$ and $\mathcal{L}_{fft}$ are $\ell_1$ loss in the RGB and Fourier domain respectively and $\lambda_{pix}, \lambda_{fft}, \lambda_{align}, \lambda_{div}, \lambda_{bal}$, and $\lambda_{con}$ are hyperparameters controlling the relative importance of each loss component.

# 4 EXPERIMENTS

## 4.1 EXPERIMENTAL SETUP

To demonstrate the effectiveness of our method, we conduct experiments mainly from the 2 aspects: (1) 6-task and 3-domain experiment, (2) 9-task and 3-domain experiment. We consider 3 image domains—natural, medical, and remote sensing images—with a diverse selection of image restoration tasks from each domain. For the 6-task setting, we include 2 tasks per domain: natural image 4× super-resolution (SR) and deraining, medical MRI SR and CT denoising, and remote sensing image 4× SR and cloud removal. For the 9-task setting, we introduce one additional task per domain: natural image motion deblurring, medical PET synthesis, and remote sensing image dehazing.

Table 1: Quantitative comparison between our method and other SOTA methods on 3 domains & 6 tasks experimental setting. The best and second-best metrics are highlighted in **bold** and underline.

| Image Domain | | Natural Image | | | | Medical Image | | | | Remote Sensing Image | | | | Average Performance | |
|---|---|---|---|---|---|---|---|---|---|---|---|---|---|---|---|
| Task & Dataset | | Super-Resolution on DIV2K-VAL | | Deraining on Rain100L | | MRI SR on IXI MRI | | CT Denoising on AAPM-Mayo | | RSI SR on UCMerced | | Cloud Removal on CUHK CR1 | | | |
| Method | Year | PSNR↑ | SSIM↑ | PSNR↑ | SSIM↑ | PSNR↑ | SSIM↑ | PSNR↑ | SSIM↑ | PSNR↑ | SSIM↑ | PSNR↑ | SSIM↑ | PSNR↑ | SSIM↑ |
| Single-Task Method | | | | | | | | | | | | | | | |
| MPRNet | CVPR2021 | 28.82 | 0.8115 | 38.07 | 0.9817 | 26.84 | 0.8891 | 33.60 | 0.9259 | 27.70 | 0.7730 | 25.35 | 0.7389 | 30.06 | 0.8534 |
| SwinIR | ICCVW2021 | 28.61 | 0.8051 | 36.07 | 0.9736 | 26.06 | 0.8766 | 33.51 | 0.9243 | 27.29 | 0.7545 | 24.36 | 0.6552 | 29.32 | 0.8482 |
| Restormer | CVPR2022 | 28.94 | 0.8158 | 38.34 | 0.9822 | 27.58 | 0.9017 | 33.69 | 0.9268 | 28.01 | 0.7844 | 25.96 | 0.7541 | 30.42 | 0.8608 |
| NAFNet | ECCV2022 | 28.73 | 0.8146 | 37.06 | 0.9773 | 27.32 | 0.8980 | 33.68 | 0.9270 | 27.87 | 0.7808 | 25.99 | 0.7591 | 30.11 | 0.8595 |
| All-in-One Method | | | | | | | | | | | | | | | |
| Transweather | CVPR2022 | 27.40 | 0.7643 | 33.20 | 0.9495 | 24.59 | 0.8181 | 31.98 | 0.9040 | 25.97 | 0.6933 | 22.95 | 0.5732 | 27.68 | 0.7837 |
| PromptIR | NeurIPS2023 | 28.77 | 0.8160 | 38.71 | 0.9831 | 27.61 | 0.9023 | 33.71 | 0.9270 | 28.05 | 0.7860 | 25.81 | 0.7518 | 30.44 | 0.8610 |
| AMIR | MICCAI2024 | 28.78 | 0.8139 | 38.10 | 0.9820 | 26.30 | 0.8793 | 33.66 | 0.9262 | 27.87 | 0.7797 | 25.75 | 0.7474 | 30.08 | 0.8548 |
| DFPIR | CVPR2025 | 27.69 | 0.7845 | 37.33 | 0.9745 | 24.59 | 0.8181 | 32.66 | 0.9137 | 25.97 | 0.6933 | 26.02 | 0.7072 | 29.04 | 0.8152 |
| AdaIR | ICLR2025 | 28.81 | 0.8157 | 38.19 | 0.9816 | 27.54 | 0.9009 | 33.68 | 0.9266 | 27.99 | 0.7840 | 26.03 | 0.7578 | 30.37 | 0.8611 |
| MoCEIR | CVPR2025 | 28.16 | 0.8156 | 38.64 | 0.9840 | 27.75 | 0.9027 | 33.74 | **0.9278** | 28.06 | 0.7843 | 26.06 | **0.7615** | 30.40 | 0.8627 |
| Multi-Domain All-in-One Method | | | | | | | | | | | | | | | |
| DATPRL-IR-**6T** (Ours) | | **28.98** | **0.8191** | 39.56 | 0.9865 | **27.88** | **0.9053** | **33.80** | 0.9278 | **28.29** | 0.7917 | **26.12** | 0.7612 | **30.77** | **0.8653** |
| DATPRL-IR-**7T** (Ours) | | 29.03 | 0.8183 | 39.65 | 0.9866 | 27.78 | 0.9037 | 33.76 | 0.9269 | 28.28 | 0.7908 | 25.91 | 0.7594 | 30.74 | 0.8643 |
| DATPRL-IR-**8T** (Ours) | | 28.99 | 0.8188 | 39.64 | 0.9866 | 27.82 | 0.9047 | 33.77 | 0.9269 | 28.31 | 0.7920 | 25.92 | 0.7590 | 30.74 | 0.8647 |
| DATPRL-IR-**9T** (Ours) | | 29.05 | 0.8181 | 39.67 | 0.9867 | 27.86 | 0.9045 | 33.77 | 0.9273 | 28.31 | 0.7913 | 26.00 | 0.7592 | 30.78 | 0.8645 |

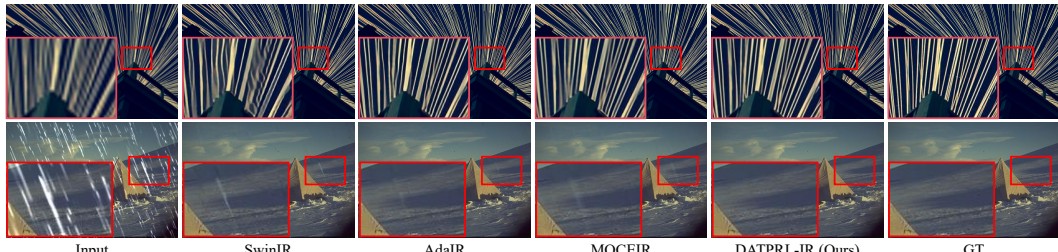

| Input | SwinIR | AdaIR | MOCEIR | DATPRL-IR (Ours) | GT |

Figure 4: Comparison of our DATPRL-IR with other SOTA methods on 6-task and 3-domain setting.

**Datasets and Evaluation Metrics.** The training datasets for each task are as follows: Natural image SR is trained on DF2K (Agustsson & Timofte, 2017; Timofte et al., 2017) dataset (DIV2K + Flickr2K) with 4× bicubic downsampling. Natural image deraining is trained using Rain100L (Yang et al., 2019). Natural image deblurring uses the GoPro (Nah et al., 2017) dataset. Following Yang et al. (2024a;b), medical MRI SR is trained on the IXI MRI dataset. Medical CT denoising uses dataset from the 2016 NIH AAPM-Mayo Clinic Low-Dose CT Grand Challenge (McCollough et al., 2017). Medical PET synthesis is trained on the PolarStar m660 dataset, where both low-quality (LQ) and high-quality (HQ) PET images are reconstructed via the standard OSEM (Hudson & Larkin, 1994) method. Remote sensing image SR is trained on the UCMerced Land Use (Yang & Newsam, 2010) dataset with 4× bicubic downsampling. Remote sensing cloud removal is trained using CUHK CR1 (Sui et al., 2024) dataset. Remote sensing dehazing is trained on RICE1 (Lin et al., 2019), which provides hazy and clean image pairs. Data augmentation including random cropping, horizontal flipping, and rotation are applied to improve robustness. Evaluation is performed on the corresponding test sets, using PSNR and SSIM (Wang et al., 2004) in RGB space as the primary metrics.

**Implementation Details.** We train our model under PyTorch (Paszke et al., 2019) framework using the Adam (Kingma, 2014) optimizer with $\beta_1 = 0.9$, $\beta_2 = 0.99$. The learning rate is initialized at $4 \times 10^{-4}$ with cosine annealing. Batch size is set to 12, and we train for 1000K iterations on NVIDIA RTX 5090 GPUs. We set the diversity threshold $\tau_{div} = 0.1$. The loss weights are set to $\lambda_{align} = 1.0$, $\lambda_{div} = 0.1$, $\lambda_{con} = 0.1$, and $\lambda_{bal} = 0.1$. For each prompt, the key is defined as a 1×1024 vector, while the value is set to 2×1024. The numbers of prompts in both the task and domain prompt pools are set to 15, with top-k selection configured as k=3 for the task prompt pool and k=5 for the domain prompt pool. The projector is a 3-layer lightweight CNN (mainly including Conv2d, AdaptiveAvgPool2d, and MLP), and the dimensionality of its output is 1024. To ensure a fair comparison, all competing methods are trained using the loss functions and specific training strategies adopted in their original papers, while all other training setting are kept the same as those used in training our model. **For more description on datasets and implementation details, please refer to Appendix C.**

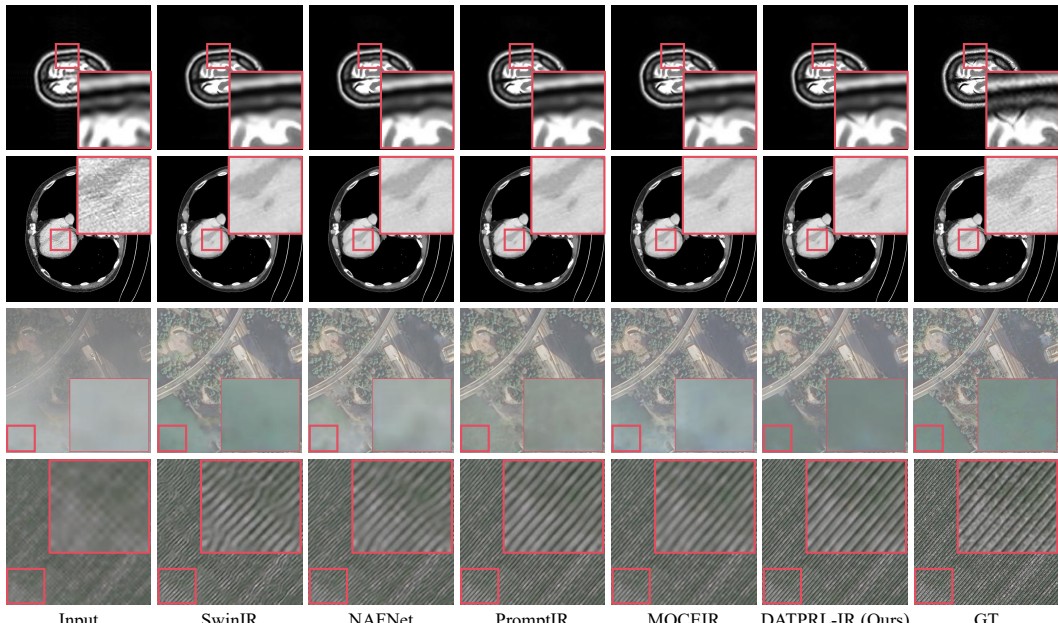

| Input | SwinIR | NAFNet | PromptIR | MOCEIR | DATPRL-IR (Ours) | GT |

Figure 5: Comparison of our DATPRL-IR with other SOTA methods on 9-task and 3-domain setting.

## 4.2 MULTI-DOMAIN ALL-IN-ONE IMAGE RESTORATION

**Results on 6-task and 3-domain all-in-one image restoration.** To validate the effectiveness of our approach, we compare it with several SOTA AiOIR methods (Zamfir et al., 2025; Cui et al., 2024; Tian et al., 2025; Yang et al., 2024a; Potlapalli et al., 2023; Valanarasu et al., 2022) and classic image restoration baselines (Chen et al., 2022; Zamir et al., 2022; Liang et al., 2021; Zamir et al., 2021). As shown in Table 1, our DATPRL-IR achieves almost comprehensive superiority across all six tasks, with an average PSNR improvement of 0.37 dB over the SOTA MoCEIR, and nearly 1 dB gain on the natural image deraining task. Furthermore, as illustrated in Figure 4, our method is able to more thoroughly remove degradations and reconstruct clearer image details compared to other methods. These results convincingly demonstrate the effectiveness of our proposed domain-aware task prompt representation learning in guiding image restoration.

**Results on 9-task and 3-domain all-in-one image restoration.** To further evaluate scalability of our DATPRL-IR, we sequentially add three tasks—natural image deblurring, PET synthesis, and remote sensing image dehazing—to train our 9-task (9T) model, while also obtaining intermediate 7-task (7T) and 8-task (8T) models. As shown in Table 1, it can be clearly observed that when the task number grows from 6 to 9, our method does not exhibit significant performance degradation on the original tasks, it even achieves a certain degree of performance improvement. This provides strong evidence for our claim that different tasks indeed share transferable knowledge that can complement each other, and our method can effectively exploit both the shared and specific knowledge to enhance model robustness when facing a larger number of tasks. As illustrated in Figure 5, images restored by our method exhibit clearer textures and fewer artifacts.

**Due to the limited space, additional quantitative and qualitative results, zero-shot and generalization results and analysis are provided in Appendix D.**

## 4.3 ABLATION STUDIES

**The effects of task prompt pool and domain prompt pool.** Table 2 investigates the impact of task prompt (TP) and domain prompt (DP) pools under 6-task and 3-domain experiment setting. Using any one type of prompt pool can provide clear improvements over the baseline, indicating that both task-aware and domain-aware representations contribute useful prior knowledge. Importantly, combining both the two pools leads to the best performance, the PSNR on the three datasets is respectively increased by 1.22dB, 0.10dB, and 0.27dB across the three tasks. The above abla-

tion results highlights the effectiveness of our dual-prompt design in enhancing generalization and robustness for scenarios with diverse tasks and domains.

Table 2: Effect of task prompt (TP) pool and domain prompt (DP) pool on DATPRL-IR.

| TP Pool | DP Pool | Deraining on Rain100L | | CT Denoising on AAPM | | RSI SR on UCMerced | |
|---|---|---|---|---|---|---|---|
| | | PSNR | SSIM | PSNR | SSIM | PSNR | SSIM |
| ✗ | ✗ | 38.34 | 0.9823 | 33.70 | 0.9269 | 28.02 | 0.7844 |
| ✓ | ✗ | 39.32 | 0.9855 | 33.76 | 0.9282 | 28.16 | 0.7901 |
| ✗ | ✓ | 38.88 | 0.9850 | 33.74 | 0.9268 | 28.12 | 0.7897 |
| ✓ | ✓ | **39.56** | **0.9865** | **33.80** | **0.9278** | **28.29** | **0.7917** |

Table 3: Effect of prompt numbers and top-k selection.

| TP Pool | | DP Pool | | 6-Task Aver. Performance | |
|---|---|---|---|---|---|
| Nums | Top K | Nums | Top K | PSNR | SSIM |
| 10 | 1 | 10 | 1 | 30.44 | 0.8607 |
| 10 | 3 | 10 | 5 | 30.53 | 0.8612 |
| 15 | 1 | 15 | 1 | 30.48 | 0.8607 |
| 15 | 3 | 15 | 5 | **30.77** | **0.8653** |
| 20 | 3 | 20 | 5 | 30.73 | 0.8635 |
| 20 | 5 | 20 | 5 | 30.70 | 0.8638 |

**The effects of prompt numbers and top-k selection.** As shown in Table 3, we present the average performance over six tasks under different configurations of prompt numbers and top-k selection. The results reveal two trends. First, enlarging the prompt pool to a moderate size improves performance by offering richer choices, while excessively large pools lead to diminishing or even negative returns, as redundant prompts may dilute useful signals. Second, for top-k selection, using too few prompts limits expressiveness, while selecting too many reduces specificity. A balanced configuration not only preserves specificity but also better leverages the shared knowledge across tasks and domains. Overall, these results indicate that a moderate prompt capacity with carefully chosen retrieval breadth are key to achieving robust and generalized image restoration performance.

**The effects of different MLLMs.** To explore our method's sensitivity to the choice of MLLMs. We test our model by replacing LLaVA1.5-7B (Liu et al., 2024) with two different MLLMs, LLaVA1.5-13B (Liu et al., 2024) and Qwen3-VL-2B-Instruct (Yang et al., 2025a; Bai et al., 2025). As shown in Table 4, the performance of our method remained stable under MLLM models of different parameter scales, with only marginal differences, demonstrating our method is not strongly dependent on a specific MLLM. This is because we only rely on relatively coarse domain-level semantics (e.g., "main image content", "shooting view", "brightness", "color" etc.), and any MLLM capable of providing such descriptions is sufficient.

**The effects of different prompt designs.** To explore the effectiveness on different prompt designs, we replace our domain prompt pool with one fixed explicit text prompt per domain (e.g., "This is an MRI image", "This is a natural image") and simpler domain encodings (one learnable $2 \times 1024$ tensor per domain). As shown in Table 5, both alternatives underperform our domain prompt pool on almost all tasks. Our method does not rely on a single fixed prompt per domain, but instead implements "shared + specific" prior modeling and learns a set of diverse domain-aware priors, allowing instance-level adaptive selection rather than forcing all MRI (or CT / RSI / Natural) images to share one identical prompt. Furthermore, using explicit prompts would require the user to know the domain beforehand, which contradicts our blind-restoration setting.

Table 4: Effectiveness on different MLLMs on 3 domains & 9 tasks experimental setting. The best metrics are highlighted in **bold**.

| Task & Dataset | Natural SR on DIV2K-Val | | Deraining on Rain100L | | CT Denoising on AAPM | | PET Synthesis on PolarStar | | RSI SR on UCMerced | | RSI Dehazing on RICE1 | |
|---|---|---|---|---|---|---|---|---|---|---|---|---|
| Method | PSNR ↑ | SSIM ↑ | PSNR ↑ | SSIM ↑ | PSNR ↑ | SSIM ↑ | PSNR ↑ | SSIM ↑ | PSNR ↑ | SSIM ↑ | PSNR ↑ | SSIM ↑ |
| LLaVA1.5-13B | 29.04 | 0.8180 | **39.71** | **0.9868** | **33.77** | 0.9272 | 37.08 | 0.9500 | **28.31** | **0.7918** | 26.92 | 0.9347 |
| LLaVA1.5-7B | 29.05 | 0.8181 | 39.67 | 0.9867 | **33.77** | **0.9273** | **37.12** | **0.9502** | **28.31** | 0.7913 | **26.94** | 0.9347 |
| Qwen3-VL-2B-Instruct | **29.07** | **0.8182** | 39.69 | **0.9868** | **33.77** | 0.9270 | 37.11 | 0.9501 | **28.31** | 0.7914 | 26.82 | **0.9351** |

Table 5: Effectiveness on different prompt designs on 3 domains & 9 tasks experimental setting. The best metrics are highlighted in **bold**.

| Task & Dataset | Deraining on Rain100L | | Deblurring on GoPro | | MRI SR on IXI MRI | | PET Synthesis on PolarStar | | Cloud Removal on CUHK CR1 | | RSI Dehazing on RICE1 | |
|---|---|---|---|---|---|---|---|---|---|---|---|---|
| Method | PSNR ↑ | SSIM ↑ | PSNR ↑ | SSIM ↑ | PSNR ↑ | SSIM ↑ | PSNR ↑ | SSIM ↑ | PSNR ↑ | SSIM ↑ | PSNR ↑ | SSIM ↑ |
| Explicit Domain Prompts | 39.59 | 0.9865 | 29.31 | 0.8839 | 27.77 | 0.9035 | 37.08 | 0.9500 | 25.76 | 0.7573 | 25.91 | 0.9296 |
| Simple Domain Encodings | 39.52 | 0.9863 | 29.21 | 0.8814 | 27.70 | 0.9031 | 37.02 | 0.9496 | 25.81 | 0.7566 | 25.81 | 0.9213 |
| Domain Prompt Pool (Ours) | **39.67** | **0.9867** | **29.57** | **0.8881** | **27.86** | **0.9045** | **37.12** | **0.9502** | **26.00** | **0.7592** | **26.94** | **0.9347** |

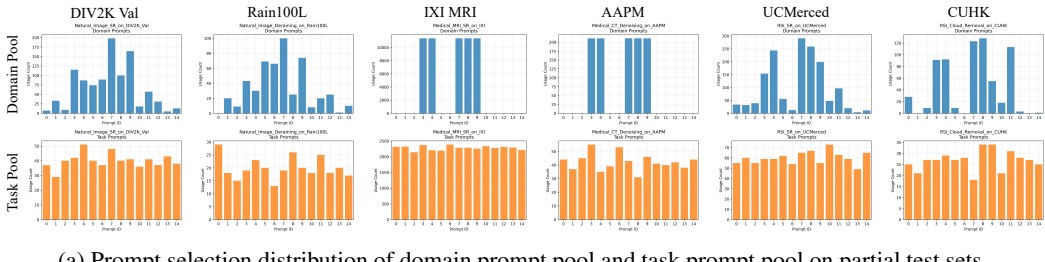

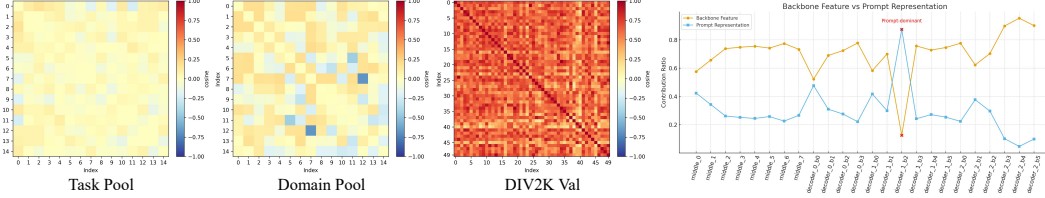

(a) Prompt selection distribution of domain prompt pool and task prompt pool on partial test sets.

(b) Diversity of the prompts in the 2 prompt pool and instance-level (c) Contribution ratio of prompt representations on DIV2K-Val. sentations and backbone features.

Figure 6: In-depth analysis of our method. Zoom in for better visualization.

## 4.4 IN-DEPTH ANALYSIS AND DISCUSSION.

**Prompt selection distribution.** As shown in Figure 6a, we visualize the selection distribution of prompts from both the domain and task prompt pools across six datasets. For the domain prompt pool, different datasets exhibit distinct distributions. Notably, medical datasets present highly uniform selections due to their relatively homogeneous image content and color patterns. In contrast, the task prompt pool shows larger overlaps across datasets, suggesting that a considerable number of task prompts are shared. This observation confirms that our method can effectively leverage shared knowledge across tasks to enhance network performance.

**Prompt diversity.** Figure 6b presents pairwise similarity heatmaps of prompt values in the two prompt pools. It is evident that each prompt has learned distinct content with little redundancy, thereby providing the model with diverse options. Furthermore, to examine instance-level behavior, we visualize the similarities between final prompt representations generated for different input images from the DIV2K validation set. For the same task, our method produces prompt representations with similar overall directions, while retaining instance-specific variations, indicating that our method enhances the instance-level diversity of learned prompt representations.

**The contribution ratio of prompt representations and backbone features.** We further analyze the learnable gating coefficient $\alpha_l \in [0, 1]$ between prompt representations and backbone features at each block from the middle layers to the decoder, which is mentioned in Sec. 3.1. As shown in Figure 6c, most blocks exhibit a dominant reliance on backbone features, indicating that the network is still primarily driven by the restoration backbone while prompt representations serve as auxiliary guidance. Furthermore, earlier blocks at each scale rely more heavily on prompt representations compared to deeper blocks, and the large variations in contribution ratios across different layers further highlight the importance of adopting adaptive fusion ratios.

## 5 CONCLUSION

In this work, we proposed the first multi-domain all-in-one image restoration (MD-AiOIR) method, DATPRL-IR, which covers multiple restoration tasks across various image domains. By introducing domain-aware task representation learning, DATPRL-IR can fully utilize both specific and shared knowledge across tasks and domains, effectively reducing the learning difficulty of the model and improving its performance. Extensive experiments show that DATPRL-IR outperforms existing SOTA methods and demonstrates excellent generalization abilities. We believe that this work lays the foundation for future research towards a more unified restoration framework.

ETHICS STATEMENT

The authors acknowledge that this work adheres to the ICLR Code of Ethics.

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

APPENDIX

## A  USE OF LARGE LANGUAGE MODELS

Large language models were used solely for light editing tasks including grammar correction, spelling checks, and minor phrasing improvements to enhance clarity and concision.

## B  DISCUSSION ON BROADER IMPLICATION AND LIMITATION

In this paper, we propose a dual-prompt-pool approach tailored to MD-AiOIR. In particular, our analysis and experiments reveal that different tasks/domains share some common knowledge, while still preserving specific differences. Modeling these 'shared + specific' priors can significantly reduce the learning difficulty when facing more tasks and domains, and help improve restoration performance. We believe this insight opens a new perspective for moving image restoration toward more unified and scalable models across multiple domains. There are also several limitations in our framework. The training cost of our framework is a little bit higher than that of single-domain and single-task models. In addition, although our method demonstrates strong zero-shot behavior, extending it to a broader range of domains remains an interesting direction. It is worth further exploring how to model the shared and different prior knowledge between various restoration tasks more efficiently and interpretably.

## C  DETAILED EXPERIMENTAL SETTING

**Datasets.** The training datasets for each task are as follows: Natural image SR is trained on the DF2K (Agustsson & Timofte, 2017; Timofte et al., 2017) dataset (DIV2K + Flickr2K) with 4× bicubic downsampling. Natural image deraining is trained using Rain100L (Yang et al., 2019). Natural image deblurring uses the GoPro (Nah et al., 2017) dataset. Following Yang et al. (2024a;b), medical MRI SR is trained on the IXI[1] MRI dataset. Medical CT denoising uses dataset from the 2016 NIH AAPM-Mayo Clinic Low-Dose CT Grand Challenge (McCollough et al., 2017). Medical PET synthesis is trained on the PolarStar m660 dataset, where both low-quality (LQ) and high-quality (HQ) PET images are reconstructed via the standard OSEM (Hudson & Larkin, 1994) method. Remote sensing image SR is trained on the UCMerced Land Use (Yang & Newsam, 2010) dataset with 4× bicubic downsampling. Remote sensing cloud removal is trained using CUHK CR1 (Sui et al., 2024) dataset. Remote sensing dehazing is trained on RICE1 (Lin et al., 2019), which provides hazy and clean image pairs. Table 6 presents the detailed numbers of training and testing images for each dataset. Data augmentation including random cropping, horizontal flipping, and rotation are applied to improve robustness. We unify all inputs by converting every dataset sample into a standard 3-channel visual image before feeding it into the network. Natural and remote-sensing datasets are already stored as RGB images. For medical datasets that are not originally stored in RGB format, we first convert them into grayscale images, and then replicate the single channel three times to obtain a 3-channel representation. All images are stored in a visual format (PNG or JPG), ensuring consistent dimensionality.

Table 6: Detailed description of the datasets utilized.

| Datasets | DF2K | Rain100L | GoPro | IXI MRI | AAPM-Mayo | PolarStar m660 | UCMerced |
|---|---|---|---|---|---|---|---|
| Train | 3450 | 200 | 2103 | 40500 | 18351 | 27837 | 1800 |
| Test | 100 | 100 | 1111 | 11400 | 211 | 2044 | 300 |
| Tasks | Natural SR | Natural Deraining | Natural Deblurring | MRI SR | CT Denoising | PET Synthesis | RSI SR |

| Datasets | CUHK CR1 | RICE1 |
|---|---|---|
| Train | 534 | 400 |
| Test | 134 | 100 |
| Tasks | Cloud Removal | RSI Dehazing |

**Implementation Details.** We train our model under PyTorch (Paszke et al., 2019) framework using the Adam (Kingma, 2014) optimizer with $\beta_1 = 0.9$, $\beta_2 = 0.99$. The learning rate is initialized

---

[1]https://brain-development.org/ixi-dataset/

Table 7: Examples of LLaVA-generated textual descriptions for medical, remote sensing, and natural images.

| **Medical Images** |
|---|
| "The image is a black and white photo of a brain, with a close up view of the frontal lobe." |
| "The image is a black and white picture of a human body, specifically focusing on the abdominal area." |
| "The image is a grayscale medical image with low overall contrast, featuring a bright background and a few small dark foci near the center." |
| **Remote Sensing Images** |
| "The image is a large, green field with a white snowy ground, and it is a winter scene." |
| "The image is a bird's-eye view of a forest with low brightness, where a river flows through it and a highway is visible nearby." |
| "The image is a bird's eye view of a large building with a pool, surrounded by palm trees." |
| **Natural Images** |
| "The image features two women in red outfits walking through a grassy field, one of them carrying a basket." |
| "The image depicts a beautiful garden entrance featuring a red door with a large vine growing up the side of the building." |
| "A colorful image taken from a boat shows tourists on red seats under a canopy looking over vivid blue sea toward a detailed coastal cityscape." |

at $4 \times 10^{-4}$ with cosine annealing. Batch size is set to 12, and we train for 1000K iterations on NVIDIA RTX 5090 GPUs. We set the diversity threshold $\tau_{\mathrm{div}} = 0.1$. The loss weights are set to $\lambda_{\mathrm{align}} = 1.0$, $\lambda_{\mathrm{div}} = 0.1$, $\lambda_{\mathrm{con}} = 0.1$, and $\lambda_{\mathrm{bal}} = 0.1$. For each prompt, the key is defined as a 1×1024 vector, while the value is set to 2×1024. The numbers of prompts in both the task and domain prompt pools are set to 15, with top-k selection configured as k=3 for the task prompt pool and k=5 for the domain prompt pool. The projector is a 3-layer lightweight CNN (mainly including Conv2d, AdaptiveAvgPool2d, and MLP), and the dimensionality of its output is 1024. The used LLaVA-v1.5-7B is primarily built from CLIP's ViT-L/14 visual encoder (Radford et al., 2021) and Vicuna-7B (Chiang et al., 2023), a language model based on the LLaMA (Touvron et al., 2023) architecture. We instruct the LLaVA-v1.5-7B to produce concise descriptions, and a very small number of texts that exceed the CLIP text encoder length limit will be automatically truncated to 77 tokens. It is worth noting that the LLaVA-v1.5-7B and the CLIP text encoder are only used during the training phase. During inference, neither the LLaVA-v1.5-7B nor CLIP is required, thereby introducing no additional inference overhead. In Table 7, we present several examples of LLaVA-generated textual descriptions for medical, remote sensing, and natural images. These descriptions provide concise and discriminative domain cues that are sufficient for alignment.

**Contrastive Regularization.** In addition to the diversity regularization and prompt entropy regularization described in the main text, we also introduce a contrastive regularization to enhance the sensitivity of instance-level prompt selection. Specifically, we adopt a contrastive objective to align the query with the keys of their most relevant prompts while pushing them away from the keys of unrelated prompts. The contrastive regularization is defined as:

$$\mathcal{L}_{\mathrm{con}} = -\log \frac{\exp(\langle \mathbf{q}, \mathbf{k}^+ \rangle / \tau)}{\exp(\langle \mathbf{q}, \mathbf{k}^+ \rangle / \tau) + \sum_{\mathbf{k}^-} \exp(\langle \mathbf{q}, \mathbf{k}^- \rangle / \tau)}, \tag{9}$$

where $\mathbf{q}$ denotes the query and $\{\mathbf{k}^+\}$ and $\{\mathbf{k}^-\}$ denote the sets of positive (selected) and negative (non-selected) prompt keys.

**Compared Methods.** We compare our DATPRL-IR with several SOTA AiOIR methods (Zamfir et al., 2025; Cui et al., 2024; Tian et al., 2025; Yang et al., 2024a; Potlapalli et al., 2023; Valanarasu et al., 2022) and classic image restoration baselines (Chen et al., 2022; Zamir et al., 2022; Liang et al., 2021; Zamir et al., 2021). To ensure a fair comparison, all compared methods are retrained from scratch on the same datasets as ours using their original loss functions and training strategies as specified in their papers. All remaining training conditions were kept identical to ours, including the train/test splits, the domain-balanced sampling strategy, data augmentations, and other shared hyperparameters.

## D  MORE EXPERIMENT RESULTS

### D.1  MULTI-DOMAIN ALL-IN-ONE IMAGE RESTORATION

We present detailed quantitative comparisons between our method and other approaches in Table 8. Our DATPRL-IR surpasses other SOTA methods across most tasks, demonstrating the superiority of our proposed domain-aware task prompt representation learning. Moreover, compared with the 6-task setting, it exhibits no performance drop on the original six tasks, but even achieves further

Table 8: Quantitative comparison between our method and other SOTA methods on 3 domains & 9 tasks experimental setting. The best and second-best metrics are highlighted in **bold** and underline.

| Image Domain | Natural Image | | | Medical Image | | | Remote Sensing Image | | |
|---|---|---|---|---|---|---|---|---|---|
| Task & Dataset | SR on DIV2K-Val | Deraining on Rain100L | Deblurring on GoPro | MRI SR on IXI MRI | CT Denoising on AAPM-Mayo | PET Synthesis on PolarStar m660 | RSI SR on UCMerced | Cloud Removal on CUHK CR1 | RSI Dehazing on RICE1 |
| **Single-Task Method** | | | | | | | | | |
| MPRNet | 28.32 / 0.8067 | 37.55 / 0.9797 | 28.02 / 0.8570 | 26.69 / 0.8871 | 33.54 / 0.9253 | 36.72 / 0.9475 | 27.47 / 0.7646 | 25.20 / 0.7334 | 25.66 / 0.9268 |
| SwinIR | 28.80 / 0.8109 | 37.55 / 0.9802 | 28.17 / 0.8510 | 26.49 / 0.8844 | 33.63 / 0.9251 | 36.78 / 0.9468 | 27.52 / 0.7670 | 25.39 / 0.7249 | 25.42 / 0.9244 |
| Restormer | 28.63 / 0.8150 | 38.45 / 0.9833 | 29.06 / 0.8805 | 27.43 / 0.8992 | 33.70 / 0.9269 | **37.20 / 0.9509** | 27.94 / 0.7827 | 25.66 / 0.7451 | 26.07 / 0.9286 |
| NAFNet | 28.64 / 0.8128 | 37.31 / 0.9783 | 29.20 / 0.8828 | 27.31 / 0.8977 | 33.66 / 0.9267 | 37.14 / 0.9505 | 27.80 / 0.7774 | 25.96 / 0.7591 | 26.45 / 0.9215 |
| **All-in-One Method** | | | | | | | | | |
| Transweather | 28.16 / 0.7951 | 32.41 / 0.9392 | 26.53 / 0.8116 | 25.55 / 0.8638 | 32.98 / 0.9209 | 36.47 / 0.9440 | 26.90 / 0.7402 | 25.35 / 0.7193 | 25.18 / 0.9226 |
| PromptIR | 28.86 / 0.8127 | 38.21 / 0.9811 | 28.79 / 0.8749 | 27.31 / 0.8964 | 33.66 / 0.9265 | 37.03 / 0.9495 | 27.91 / 0.7809 | 25.33 / 0.7360 | 26.86 / **0.9399** |
| AdaIR | 28.24 / 0.8153 | 38.40 / 0.9834 | 29.21 / 0.8835 | 27.52 / 0.9013 | 33.70 / 0.9270 | 37.17 / 0.9508 | 27.96 / 0.7837 | 25.40 / 0.7274 | 25.40 / 0.9274 |
| MoCEIR | 28.68 / 0.8152 | 38.26 / 0.9827 | 29.32 / 0.8855 | 27.62 / 0.9003 | 33.72 / **0.9278** | 37.16 / 0.9502 | 28.00 / 0.7809 | 25.83 / 0.7586 | 26.31 / 0.9395 |
| **Muti-Domain All-in-One Method** | | | | | | | | | |
| DATPRL-IR-**9T** (Ours) | **29.05 / 0.8181** | **39.67 / 0.9867** | **29.57 / 0.8881** | **27.86 / 0.9045** | **33.77** / 0.9273 | 37.12 / 0.9502 | **28.31 / 0.7913** | **26.00** / 0.7592 | **26.94** / 0.9347 |

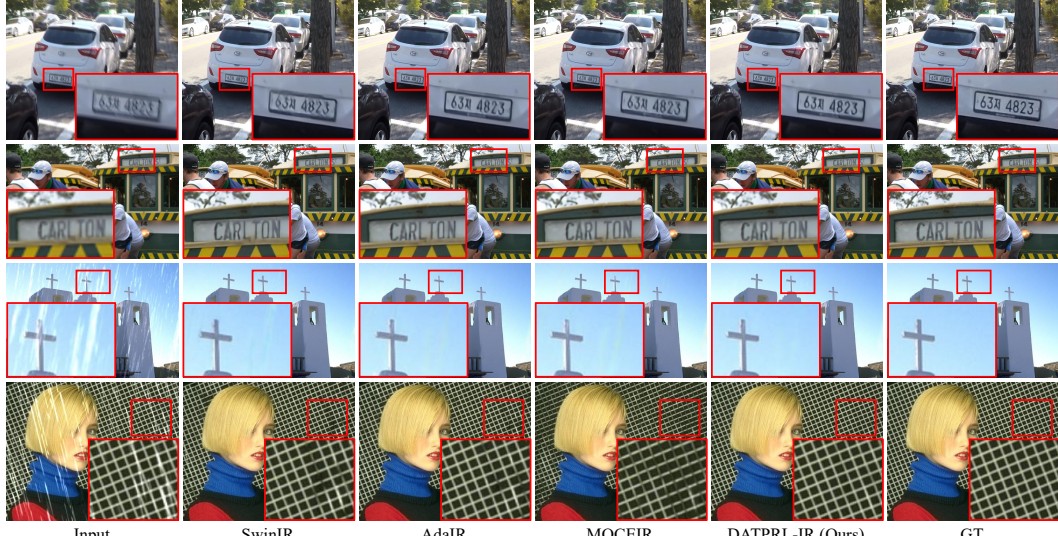

| Input | SwinIR | AdaIR | MOCEIR | DATPRL-IR (Ours) | GT |

Figure 7: Comparison of our DATPRL-IR with other SOTA methods on Natural Images.

improvements. We also provide additional qualitative visual comparisons in Figure 7, Figure 8, and Figure 9. Obviously, compared to other methods, our method is able to remove degradations clearer and reconstruct more image details. These results significant demonstrate the effectiveness of our method.

## D.2 ZERO-SHOT AND GENERALIZATION PERFORMANCE

To demonstrate the zero-shot and generalization capability of our model, we add another three evaluation settings: (1) Zero-shot on unseen image domains; (2) Generalization to unseen distributions; (3) Zero-shot on completely unseen restoration tasks. All experiments are performed using the well-trained model, and no fine-tuning is applied to any method.

**Zero-shot on unseen image domains.** We test zero-shot super-resolution (SR) performance on two image domains never seen during training: AI-generated content (AIGC) images and comic images. Following (Zuo et al., 2025), we use the standard AIGC benchmark GenAI-Bench (Li et al., 2024) for testing. Specifically, we randomly selected 100 images from each of the three subsets SDXL-Turbo, DeepFloyd, and Midjourney, and performed 4x bicubic downsampling to construct our AIGC SR test set. For comic image SR, we chose the most commonly used Manga109(Matsui et al., 2017) as the test set. As shown in Table 9, our method comprehensively outperforms other approaches in two unseen image domains, achieving PSNR improvements of 0.87dB and 0.45dB over the prompt-based PromptIR on SDXL-Turbo and Manga109, respectively. Furthermore, the visual comparison in Figure 10 demonstrates that our method can reconstruct much clearer texture details, with fewer blurry edges and artifacts.

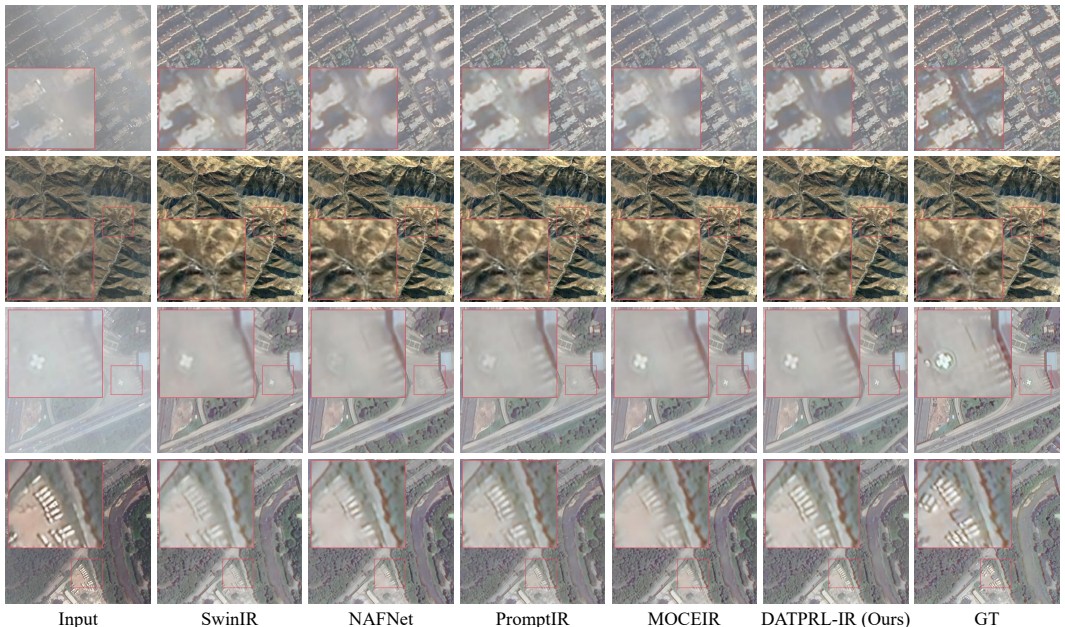

| Input | SwinIR | NAFNet | PromptIR | MOCEIR | DATPRL-IR (Ours) | GT |

Figure 8: Comparison of our DATPRL-IR with other SOTA methods on Remote Sensing Images.

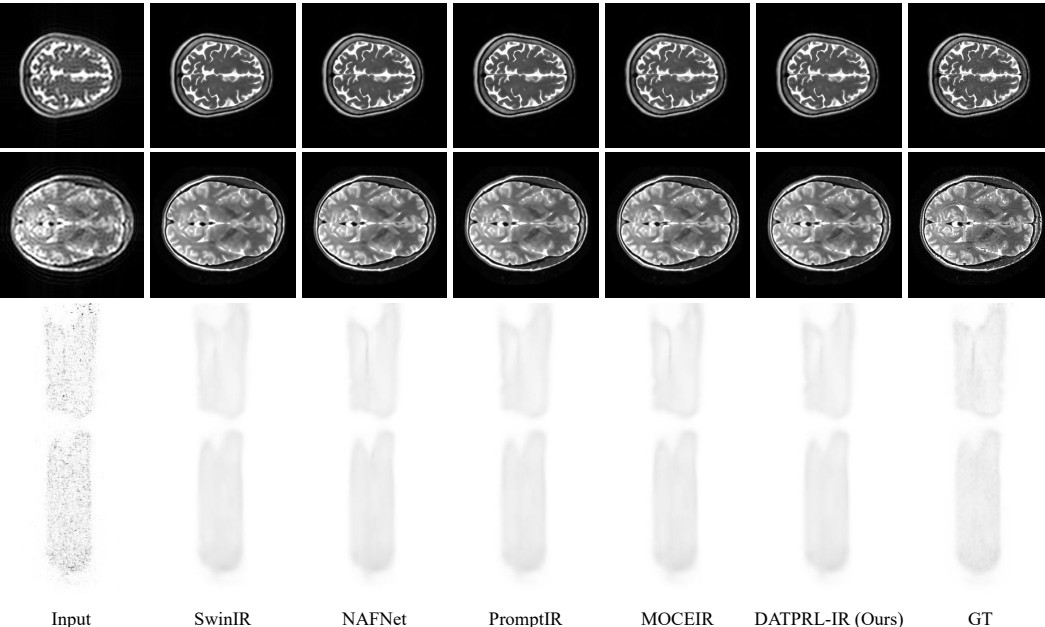

| Input | SwinIR | NAFNet | PromptIR | MOCEIR | DATPRL-IR (Ours) | GT |

Figure 9: Comparison of our DATPRL-IR with other SOTA methods on Medical Images.

**Zero-shot on completely unseen restoration tasks.** We further test zero-shot performance on 2 tasks not included in training: natural image desnowing, and specular highlight removal. For image desnowing, following (Lin et al., 2025), we randomly select 2,000 pairs from the Snow100K-S (Liu et al., 2018) test subset for evaluation. For specular highlight removal, we evaluate all methods on the large-scale real-world benchmark SHIQ (Fu et al., 2021), which contains 1000 real image pairs for testing. As shown in Table 10, our method surpasses the SOTA MoCEIR by 0.92 dB and 0.55 dB in PSNR on specular highlight removal and image desnowing tasks respectively, without any fine-tuning. At the same time, Figure 11 and Figure 12 show the comparison of the visual results of our method with other SOTA methods on these two completely unseen tasks. Our approach demonstrates extremely strong zero-shot capability, especially in specular highlight removal, where

Table 9: Quantitative zero-shot super-resolution comparison between our method and other SOTA methods on AIGC and comic images. The best metrics are highlighted in **bold**.

| Image Domain & Task | | AI-Generated Content (AIGC) SR | | | | | | Comic Image SR | |
|---|---|---|---|---|---|---|---|---|---|
| Dataset | | on SDXL-Turbo | | on DeepFloyd | | on Midjourney | | on Manga109 | |
| Method | Year | PSNR ↑ | SSIM ↑ | PSNR ↑ | SSIM ↑ | PSNR ↑ | SSIM ↑ | PSNR ↑ | SSIM ↑ |
| Single-Task Method | | | | | | | | | |
| MPRNet | CVPR2021 | 31.13 | 0.8730 | 26.76 | 0.8254 | 27.19 | 0.7656 | 27.13 | 0.8612 |
| SwinIR | ICCVW2021 | 31.20 | 0.8709 | 27.41 | 0.8344 | 27.19 | 0.7627 | 27.76 | 0.8697 |
| Restormer | CVPR2022 | 32.95 | 0.8985 | 28.81 | 0.8675 | 28.24 | 0.8005 | 28.13 | 0.8758 |
| NAFNet | ECCV2022 | 32.67 | 0.8959 | 28.59 | 0.8637 | 28.12 | 0.7965 | 27.98 | 0.8727 |
| All-in-One Method | | | | | | | | | |
| Transweather | CVPR2022 | 32.49 | 0.8916 | 27.73 | 0.8456 | 27.81 | 0.7866 | 26.00 | 0.8330 |
| PromptIR | NeurIPS2023 | 32.38 | 0.8933 | 28.56 | 0.8642 | 27.90 | 0.7916 | 28.28 | 0.8756 |
| AdaIR | ICLR2025 | 32.95 | 0.8989 | 28.87 | 0.8687 | 28.23 | 0.8008 | 28.05 | 0.8758 |
| MoCEIR | CVPR2025 | 33.00 | 0.8995 | 28.90 | 0.8680 | 28.30 | 0.8027 | 28.10 | 0.8745 |
| Muti-Domain All-in-One Method | | | | | | | | | |
| DATPRL-IR (Ours) | | **33.25** | **0.9017** | **29.07** | **0.8713** | **28.38** | **0.8045** | **28.73** | **0.8828** |

Table 10: Quantitative zero-shot comparison between our method and other SOTA methods on unseen image specular highlight removal, and image desnowing. The best metrics are highlighted in **bold**.

| Task & Dataset | | Highlight Removal on SHIQ | | Image Desnowing on Snow100K-S | |
|---|---|---|---|---|---|
| Method | Year | PSNR ↑ | SSIM ↑ | PSNR ↑ | SSIM ↑ |
| Single-Task Method | | | | | |
| MPRNet | CVPR2021 | 25.83 | 0.9297 | 26.09 | 0.8882 |
| Restormer | CVPR2022 | 25.67 | 0.9295 | 26.57 | 0.8955 |
| NAFNet | ECCV2022 | 25.89 | 0.9298 | 26.27 | 0.8923 |
| All-in-One Method | | | | | |
| Transweather | CVPR2022 | 26.20 | 0.9315 | 26.36 | 0.8870 |
| PromptIR | NeurIPS2023 | 24.87 | 0.9239 | 26.42 | 0.8923 |
| AdaIR | ICLR2025 | 25.35 | 0.9263 | 26.54 | 0.8951 |
| MoCEIR | CVPR2025 | 25.30 | 0.9259 | 26.65 | 0.8976 |
| Muti-Domain All-in-One Method | | | | | |
| DATPRL-IR (Ours) | | **26.22** | **0.9336** | **27.20** | **0.9018** |

our method can accurately remove degradation and reconstruct lost textures, while other methods can hardly remove any highlights.

**Generalization to unseen distributions.** We evaluate generalization performance on real-world image deraining and remote-sensing image (RSI) deblurring, both of which come from distributions different from the training data. Real-world image deraining performance is tested on SPA-Data (Wang et al., 2019), which has 1000 pairs of real-world test images. As for RSI deblurring, we randomly add varying degrees of Gaussian blur, defocus blur, and motion blur to the UCMerced (Yang & Newsam, 2010) test set, creating 300 pairs of clean-blurred remote sensing images for testing. As shown in Table 11, our method comprehensively outperforms other AiOIR methods on both tasks. In particular, on the RSI deblurring task, it achieves an almost 2dB PSNR advantage over the SOTA methods MoCEIR, AdaIR, and PromptIR. In addition, as shown in Figure 13, the comparative methods struggle to effectively remove blur from images and have issues with task recognition errors, resulting in restored images with significant color shifts. In contrast, our method can effectively remove blur while preserving colors and other image content.

The above experiments demonstrate the two learned prompt pools capture reusable domain and task priors rather than dataset-specific patterns. When tested on unseen domains, tasks, and distributions, our model can adapt by selecting and composing prompts that best match the new data distribution, demonstrating clear transferability. This further proves the importance of modeling both shared knowledge and unique knowledge across different tasks/domains for image restoration.

Table 11: Quantitative generalization comparison between our method and other SOTA methods on real-world image deraining and RSI deblurring. The best metrics are highlighted in **bold**.

| Task & Dataset | | Real-World Deraining on SPA-Data | | RSI Deblurring on UCMerced | |
|---|---|---|---|---|---|
| Method | Year | PSNR ↑ | SSIM ↑ | PSNR ↑ | SSIM ↑ |
| Single-Task Method | | | | | |
| MPRNet | CVPR2021 | 31.99 | 0.9263 | 26.70 | 0.7741 |
| Restormer | CVPR2022 | **32.35** | **0.9282** | 27.21 | 0.7928 |
| NAFNet | ECCV2022 | 31.99 | 0.9249 | 26.92 | 0.7864 |
| All-in-One Method | | | | | |
| Transweather | CVPR2022 | 32.21 | 0.9304 | 26.60 | 0.7684 |
| PromptIR | NeurIPS2023 | 32.09 | 0.9254 | 26.77 | 0.7828 |
| AdaIR | ICLR2025 | 32.28 | 0.9278 | 27.22 | 0.7951 |
| MoCEIR | CVPR2025 | 32.20 | 0.9261 | 27.35 | 0.7956 |
| Muti-Domain All-in-One Method | | | | | |
| DATPRL-IR (Ours) | | 32.32 | 0.9265 | **28.91** | **0.8108** |

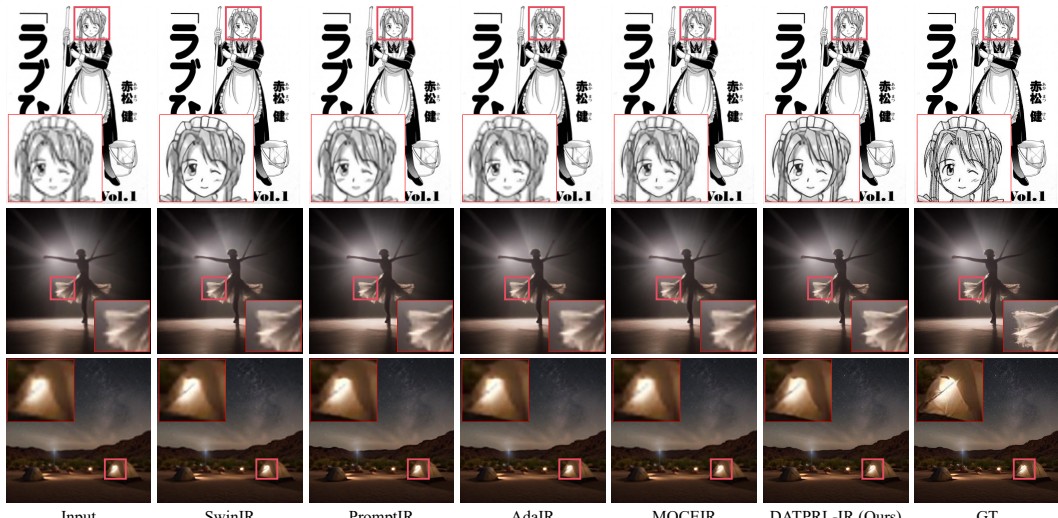

| Input | SwinIR | PromptIR | AdaIR | MOCEIR | DATPRL-IR (Ours) | GT |

Figure 10: Zero-shot comparison of our DATPRL-IR with other SOTA methods on AIGC and comic domains. Please zoom in for better visualization.

### D.3 PERCEPTUAL QUALITY COMPARISON

For more comprehensive evaluation, as shown in Table 12, we test five perceptual metrics—CLIPIQA (Wang et al., 2023b), MANIQA (Yang et al., 2022), MUSIQ (Ke et al., 2021), NIQE (Zhang et al., 2015), and FID (Heusel et al., 2017)—under the 3-domain and 9-task setting. Across all the three tasks, our DATPRL-IR consistently achieves the best scores in CLIPIQA and FID. In image deraining task, our method almost entirely outperforms other methods, achieving the best results on four metrics: CLIPIQA, MANIQA, and MUSIQ, and FID. Overall, our method achieves better or comparable perceptual scores compared to existing methods. It worth noting that to ensure fair and meaningful evaluation, we report these metrics only on natural-image tasks, because CLIPIQA, MANIQA, and MUSIQ, etc. are pre-trained exclusively on natural-image datasets, and applying them to medical or remote-sensing images would not provide reliable or interpretable scores. Moreover, medical and remote-sensing communities (Chen et al., 2025; Yang et al., 2024b;a; Lei et al., 2017; Wu et al., 2023) as well as AiOIR community (Potlapalli et al., 2023; Zamfir et al., 2025; Cui et al., 2024; Tian et al., 2025; Li et al., 2022) primarily focus more on fidelity-based (pixel-level) metrics due to their optimization objective and safety requirements.

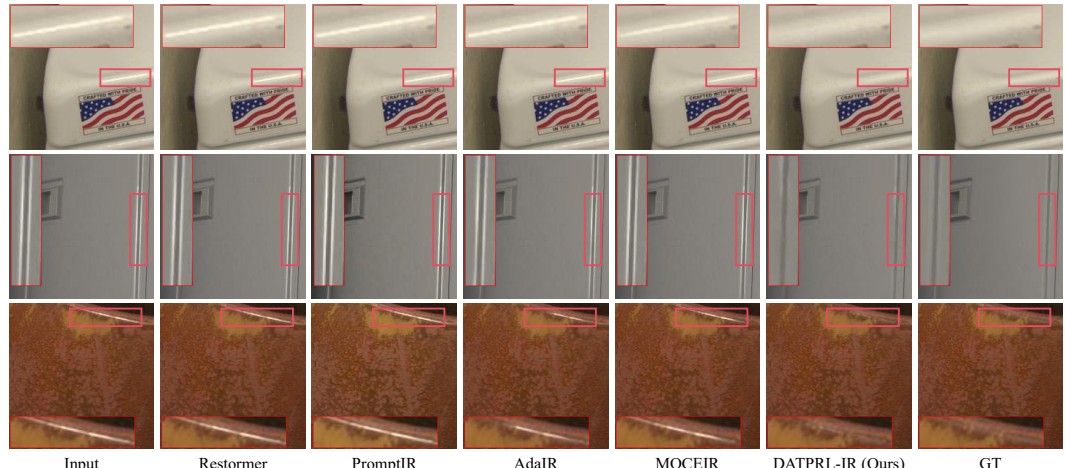

| Input | Restormer | PromptIR | AdaIR | MOCEIR | DATPRL-IR (Ours) | GT |

Figure 11: Zero-shot comparison of our DATPRL-IR with other SOTA methods on specular highlight removal task. Please zoom in for better visualization.

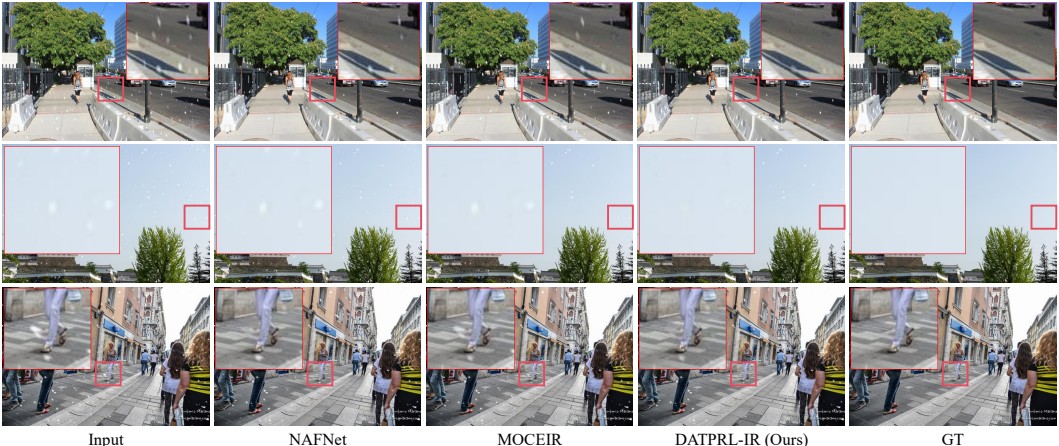

| Input | NAFNet | MOCEIR | DATPRL-IR (Ours) | GT |

Figure 12: Zero-shot comparison of our DATPRL-IR with other SOTA methods on image desnowing task. Please zoom in for better visualization.

## E   MORE ABLATION STUDIES

**Effectiveness on different fusion strategies.** To evaluate the effectiveness on different fusion strategies, except for our fusion strategy where the task prompt representation is combined with the domain prompt representation and then further fused with the image features in the decoder layers, we implement another reasonable fusion strategy, domain→image→task sequential fusion variant, for

Table 12: Perceptual quality comparison between our method and other SOTA methods on 3 domains & 9 tasks experimental setting. The best metrics are highlighted in **bold**

| Image Domain | | \multicolumn{15}{c}{Natural Image} | | | | | | | | | | | | | |
|---|---|---|---|---|---|---|---|---|---|---|---|---|---|---|---|---|
| Task & **Dataset** | | \multicolumn{5}{c}{SR on DIV2K-Val} | | | | | \multicolumn{5}{c}{Deraining on Rain100L} | | | | | \multicolumn{5}{c}{Deblurring on GoPro} | | | | |
| Method | Year | CLIPIQA↑ | MANIQA↑ | MUSIQ↑ | NIQE↓ | FID↓ | CLIPIQA↑ | MANIQA↑ | MUSIQ↑ | NIQE↓ | FID↓ | CLIPIQA↑ | MANIQA↑ | MUSIQ↑ | NIQE↓ | FID↓ |
| Single-Task Method | | | | | | | | | | | | | | | | |
| MPRNet | CVPR2021 | 0.4561 | 0.4998 | 55.35 | 5.9716 | 104.76 | 0.7480 | 0.6967 | 70.61 | **3.1747** | 9.0289 | 0.3218 | 0.4116 | 34.01 | 5.6452 | 20.73 |
| SwinIR | ICCVW2021 | 0.4512 | 0.5059 | 58.06 | 5.9690 | 102.31 | 0.7560 | 0.6987 | 70.63 | 3.1858 | 8.3961 | 0.3142 | 0.4068 | 35.91 | 5.6113 | 25.26 |
| Restormer | CVPR2022 | 0.4634 | 0.5138 | **58.28** | 6.0288 | 102.01 | 0.7543 | 0.7010 | 70.85 | 3.2066 | 6.5893 | 0.3468 | 0.4429 | 36.96 | 5.5778 | 17.05 |
| NAFNet | ECCV2022 | 0.4607 | 0.5041 | 56.10 | **5.9210** | 102.45 | 0.7530 | 0.6976 | 70.68 | 3.2944 | 8.9292 | 0.3410 | 0.4410 | 37.24 | 5.4487 | 16.77 |
| All-in-One Method | | | | | | | | | | | | | | | | |
| Transweather | CVPR2022 | 0.4500 | 0.4828 | 51.85 | 6.6003 | 102.20 | 0.6803 | 0.6682 | 69.07 | 3.4578 | 33.030 | 0.3023 | 0.3472 | 27.81 | 6.1708 | 36.66 |
| PromptIR | NeurIPS2023 | 0.4585 | 0.5093 | 56.48 | 6.2211 | 101.95 | 0.7545 | 0.6998 | 70.81 | 3.2141 | 7.5467 | 0.3385 | 0.4382 | 36.39 | 5.6221 | 18.25 |
| AdaIR | ICLR2025 | 0.4624 | **0.5161** | 58.18 | 6.0680 | 102.18 | 0.7556 | 0.7010 | 70.86 | 3.2149 | 6.8281 | 0.3201 | 0.4547 | **38.07** | 5.5143 | 16.79 |
| MoCEIR | CVPR2025 | 0.4604 | 0.5120 | 55.47 | 6.0852 | 102.98 | 0.7573 | 0.7011 | 70.85 | 3.2167 | 7.1322 | 0.3270 | **0.4597** | 37.87 | 5.5175 | 16.88 |
| Muti-Domain All-in-One Method | | | | | | | | | | | | | | | | |
| DATPRL-IR  (Ours) | | **0.5558** | 0.5156 | 56.29 | 6.2490 | **101.93** | **0.7587** | **0.7033** | **70.94** | 3.2410 | **5.1064** | **0.3485** | 0.4461 | 35.93 | **5.4463** | **16.73** |

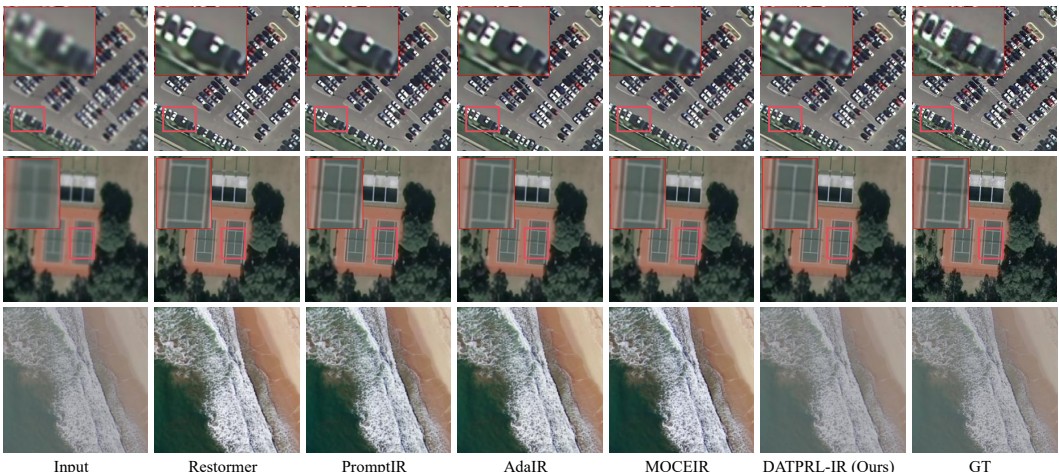

| Input | Restormer | PromptIR | AdaIR | MOCEIR | DATPRL-IR (Ours) | GT |

Figure 13: Generalization comparison of our DATPRL-IR with other SOTA methods on RSI deblurring task. Please zoom in for better visualization.

Table 13: Effectiveness on different fusion strategies on 3 domains & 9 tasks experimental setting. The best metrics are highlighted in **bold**.

| Task & Dataset | Deraining on Rain100L | | Deblurring on GoPro | | CT Denoising on AAPM | | RSI SR on UCMerced | | Cloud Removal on CUHK CR1 | | RSI Dehazing on RICE1 | |
|---|---|---|---|---|---|---|---|---|---|---|---|---|
| Method | PSNR ↑ | SSIM ↑ | PSNR ↑ | SSIM ↑ | PSNR ↑ | SSIM ↑ | PSNR ↑ | SSIM ↑ | PSNR ↑ | SSIM ↑ | PSNR ↑ | SSIM ↑ |
| Sequential Fusion (domain→image→task) | 39.60 | 0.9865 | 29.30 | 0.8832 | 33.76 | 0.9271 | 28.27 | 0.7902 | 25.82 | 0.7577 | 26.76 | 0.9376 |
| Ours | **39.67** | **0.9867** | **29.57** | **0.8881** | **33.77** | **0.9273** | **28.31** | **0.7913** | **26.00** | **0.7592** | **26.94** | **0.9347** |

comparison. We use cross-attention mechanisms to sequentially fuse domain prompt representation with image features, and then fuse the combined features with task prompt representation. As shown in Table 13, although sequential fusion also achieved considerable performance, its performance still declined compared to our direct fusion. In MD-AiOIR, both domain prompt representation and task prompt representation are equally important. Our fusion method aims to learn a prompt representation that is both task-aware and domain-aware to guide the restoration network. In addition, compared to sequential fusion, our fusion method can also use the proposed adaptive gated fusion (AGF) to control the contribution ratio between backbone features and domain-aware task prompt representation, which is also very important.

