# OpenReview forum: "Learning Domain-Aware Task Prompt Representations for Multi-Domain All-in-One Image Restoration"
_ICLR.cc/2026/Conference — ICLR 2026 Poster_

### Official Review · Reviewer_mrLL · 2025-10-27

**Soundness:** 3
**Presentation:** 3
**Contribution:** 3
**Rating:** 6
**Confidence:** 4

**Summary:**

This paper proposes **DATPRL-IR**, the first method for **Multi-Domain All-in-One Image Restoration (MD-AiOIR)**, which unifies diverse restoration tasks across natural, medical, and remote sensing image domains within a single model. The core idea is **Domain-Aware Task Prompt Representation Learning (DATPRL)**, which leverages two prompt pools: one for tasks (e.g., deraining, super-resolution) and one for domains (e.g., medical, remote sensing). Task prompts are learned end-to-end, while domain prompts are distilled from textual descriptions generated by a multimodal large language model (LLaVA) and aligned via CLIP. These prompts are adaptively selected and composed into instance-level representations, then fused via cross-attention and adaptive gating to guide the restoration backbone. Experiments on 6-task and 9-task settings across three domains show consistent improvements over state-of-the-art (SOTA) methods in PSNR and SSIM, with strong generalization and scalability.

**Strengths:**

- **Novel problem setting**: MD-AiOIR is a meaningful extension beyond single-domain AiOIR.
- **Thoughtful architecture**: The dual-prompt-pool design with adaptive composition and fusion is well-motivated.
- **Comprehensive experiments**: Evaluation across three domains and up to nine tasks is thorough, with solid PSNR/SSIM gains over SOTA.

**Weaknesses:**

**Insufficient description of task prompt learning**: The paper states that the task prompt pool consists of key-value pairs $(K_j^{\text{task}}, V_j^{\text{task}})$ and that they are “optimized jointly with restoration objectives,” but it **does not clarify**:
  - How the prompts are **initialized** (random? pre-trained?).
  - Whether keys and values are **learned independently** or share structure.
  - The **dimensionality and architecture** of the projector that maps features to the query.
  - How the **value tensor** $V_j^{\text{task}} \in \mathbb{R}^{T \times d}$ is used in the restoration backbone (e.g., is it injected via cross-attention? added as bias?).

  Without these details, the **reproducibility and technical novelty** of the task prompt mechanism are hard to assess. In contrast, the domain prompt distillation pipeline is clearly explained.

- **Limited gain from domain prompts**: As shown in Table 2, the domain prompt pool contributes only marginal improvements (e.g., +0.09 dB on CT denoising), raising questions about its necessity. Simpler domain encoding (e.g., one-hot vectors) might suffice.
- **Reliability of LLaVA-generated descriptions**: Medical and remote sensing images often lack rich semantics. It’s unclear whether LLaVA can reliably generate meaningful, discriminative textual descriptions for such domains (e.g., “grayscale + human organs” may be too vague).
- **Unverified zero-shot claims**: The abstract and conclusion mention “strong zero-shot capabilities,” but no experiments on unseen domains or tasks are provided to substantiate this.
- **Minor typo**: The method is referred to as “DATRL-IR” in one sentence of the abstract, missing the “P”.

**Questions:**

1. Could you clarify how the task prompts (keys and values) are initialized and updated during training? Are they fully learnable parameters?
2. How is the task prompt representation $PR_t$ actually **integrated into the restoration network**? Is it used as a conditioning signal in attention layers, or elsewhere?
3. Given the small performance gain from domain prompts, have you considered simpler domain encodings (e.g., one-hot vectors) as a baseline?
4. Can you provide example LLaVA-generated captions for medical/remote sensing images? How diverse and informative are they?
5. Have you compared your domain prompt pool against simpler alternatives (e.g., learnable domain embeddings)? Is the performance gain worth the added complexity?
6. The paper claims “zero-shot capabilities”—can you show results on an unseen domain (e.g., train on natural + medical, test on remote sensing)?
7. How sensitive is performance to the number of prompts (N=15) and top-k values (k=3/5)? Was this tuned extensively?

---

> ### Author Response · Authors · 2025-11-27
>
> > Insufficient description of task prompt learning. [...]
>
> Many thanks for pointing this out. We apologize for the lack of clarity and have revised the manuscript accordingly. Below is a clearer summary of the implementation details:
>
> - All task-prompt keys and values are initialized with random weights and treated as fully learnable parameters.
> - The keys and values are learned independently.
> - The projector is a 3-layer lightweight CNN (mainly including Conv2d, AdaptiveAvgPool2d, and MLP), and the dimensionality of its output is 1024.
> - Several value tensors in task prompt pool are selected and combined into a task prompt representation using the proposed PCM module. This task representation is then fused with the domain prompt representation via cross-attention to obtain a domain-aware task prompt. Finally, this domain-aware task prompt is injected into each decoder block as a conditioning signal through the proposed adaptive gated fusion (AGF) module. Please refer to Sections 3.1 and 3.2 for more information.
>
> These implementation details have been added to the revised manuscript.
>
> > Limited gain from domain prompts [...]. have you considered simpler domain encodings [...].
>
> Thanks for the thoughtful comment.
>
> - Although the gain on CT denoising is small—mainly because this task has very limited room for further improvement—the benefits are evident on other tasks. For example, adding domain prompts yields a 0.54 dB increase on deraining and improves the average PSNR across all tasks by 0.22 dB.
>
> - As suggested, we conducted experiments replacing the domain prompt pool with simpler domain encodings (one learnable tensor per domain), keeping all other settings unchanged. In addition, as raised by Reviewer 8geW, we also evaluated simpler text descriptions (e.g., “This is an MRI image”, “This is a natural image”, etc.). Results in the table below show that both alternatives underperform the domain prompt pool across all tasks. We have included these results in the revised manuscript. (Please see Appendix E for more detialed results and analyses.)
>
> **Table: Effectiveness of Different Prompt Designs on 3 Domains & 9 Tasks. Best results are in **bold**.**
>
> | Method | Deraining (Rain100L) | Deblurring (GoPro) | MRI SR (IXI MRI) | PET Synthesis (PolarStar) | Cloud Removal (CUHK CR1) | RSI Dehazing (RICE1) |
> |--------|------------------------|----------------------|---------------------|-----------------------------|-----------------------------|-------------------------|
> | Explicit Domain Prompts | 39.59 / 0.9865 | 29.31 / 0.8839 | 27.77 / 0.9035 | 37.08 / 0.9500 | 25.76 / 0.7573 | 25.91 / 0.9296 |
> | Simple Domain Encodings | 39.52 / 0.9863 | 29.21 / 0.8814 | 27.70 / 0.9031 | 37.02 / 0.9496 | 25.81 / 0.7566 | 25.81 / 0.9213 |
> | **Prompt Pool (Ours)** | **39.67 / 0.9867** | **29.57 / 0.8881** | **27.86 / 0.9045** | **37.12 / 0.9502** | **26.00 / 0.7592** | **26.94 / 0.9347** |

---

> ### Author Response · Authors · 2025-11-27
>
> > Reliability of LLaVA-generated descriptions: [...] Can you provide example LLaVA-generated captions for medical/remote sensing images? How diverse and informative are they?
>
> We appreciate the questions. We instruct LLaVA1.5-7B to output concise descriptions. In the Table below, we present several examples of LLaVA-generated textual descriptions for medical, remote sensing, and natural images.
>
> **Table: Examples of LLaVA-generated textual descriptions.**
>
> | **Medical Images** |
> |--------------------|
> | “The image is a black and white photo of a brain, with a close up view of the frontal lobe.” |
> | “The image is a black and white picture of a human body, specifically focusing on the abdominal area.” |
> | “The image is a grayscale medical image with low overall contrast, featuring a bright background and a few small dark foci near the center.” |
>
> | **Remote Sensing Images** |
> |---------------------------|
> | “The image is a large, green field with a white snowy ground, and it is a winter scene.” |
> | “The image is a bird's-eye view of a forest with low brightness, where a river flows through it and a highway is visible nearby.” |
> | “The image is a bird's eye view of a large building with a pool, surrounded by palm trees.” |
>
> | **Natural Images** |
> |---------------------|
> | “The image features two women in red outfits walking through a grassy field, one of them carrying a basket.” |
> | “The image depicts a beautiful garden entrance featuring a red door with a large vine growing up the side of the building.” |
> | “A colorful image taken from a boat shows tourists on red seats under a canopy looking over vivid blue sea toward a detailed coastal cityscape.” |
>
> These descriptions provide concise but discriminative domain cues that are sufficient for alignment. As shown in our t-SNE visualization (please see Figure 3), the resulting text features form clear domain clusters while also capturing inter-domain commonalities. This supports the motivation for learning a domain prompt pool that models both shared and specific domain characteristics. We have added these example descriptions in the revised manuscript (please see Table5 in Appendix C).

---

> ### Author Response · Authors · 2025-11-27
>
> > Unverified zero-shot claims: [...]
>
> We apologize for the missing of this important experiment. To comprehensively demonstrate the zero-shot and generalization capabilities of our model, we have added three new types of evaluations in the revised version (Please see Appendix D.2). All experiments use the well-trained models with no fine-tuning applied. Please refer to Tables 6–8 and Figures 10–13 in the Appendix for detailed quantitative results and visual comparisons.
>
> **1. Zero-shot on unseen domains**: we test performance on two domains not seen during training: AIGC images and comic images.
>
> **Table: Quantitative zero-shot SR comparison on AIGC and Comic domains. Best metrics are in bold.**
>
> | | SDXL-Turbo | DeepFloyd | Midjourney | Manga109 |
> |--------|------------|-----------|------------|----------|
> | Method | PSNR / SSIM | PSNR / SSIM | PSNR / SSIM | PSNR / SSIM |
> | **Single-Task Method** |||||
> | MPRNet | 31.13 / 0.8730 | 26.76 / 0.8254 | 27.19 / 0.7656 | 27.13 / 0.8612 |
> | SwinIR | 31.20 / 0.8709 | 27.41 / 0.8344 | 27.19 / 0.7627 | 27.76 / 0.8697 |
> | Restormer | 32.95 / 0.8985 | 28.81 / 0.8675 | 28.24 / 0.8005 | 28.13 / 0.8758 |
> | NAFNet | 32.67 / 0.8959 | 28.59 / 0.8637 | 28.12 / 0.7965 | 27.98 / 0.8727 |
> | **All-in-One Method** |||||
> | Transweather | 32.49 / 0.8916 | 27.73 / 0.8456 | 27.81 / 0.7866 | 26.00 / 0.8330 |
> | PromptIR | 32.38 / 0.8933 | 28.56 / 0.8642 | 27.90 / 0.7916 | 28.28 / 0.8756 |
> | AdaIR | 32.95 / 0.8989 | 28.87 / 0.8687 | 28.23 / 0.8008 | 28.05 / 0.8758 |
> | MoCEIR | 33.00 / 0.8995 | 28.90 / 0.8680 | 28.30 / 0.8027 | 28.10 / 0.8745 |
> | **Multi-Domain All-in-One Method** |||||
> | **DATPRL-IR (Ours)** | **33.25 / 0.9017** | **29.07 / 0.8713** | **28.38 / 0.8045** | **28.73 / 0.8828** |
>
> **2. Zero-shot on completely unseen tasks**: we further test the models on two tasks not included in training: natural image desnowing and specular highlight removal.
>
> **Table: Quantitative zero-shot comparison on specular highlight removal and desnowing  .**
>
> |  | SHIQ (Highlight Removal) | Snow100K-S (Desnowing) |
> |--------|----------------------------|--------------------------|
> | Method | PSNR / SSIM | PSNR / SSIM | PSNR / SSIM | PSNR / SSIM |
> | **Single-Task Method** |||
> | MPRNet | 25.83 / 0.9297 | 26.09 / 0.8882 |
> | Restormer | 25.67 / 0.9295 | 26.57 / 0.8955 |
> | NAFNet | 25.89 / 0.9298 | 26.27 / 0.8923 |
> | **All-in-One Method** |||
> | Transweather | 26.20 / 0.9315 | 26.36 / 0.8870 |
> | PromptIR | 24.87 / 0.9239 | 26.42 / 0.8923 |
> | AdaIR | 25.35 / 0.9263 | 26.54 / 0.8951 |
> | MoCEIR | 25.30 / 0.9259 | 26.65 / 0.8976 |
> | **Multi-Domain All-in-One Method** |||
> | **DATPRL-IR (Ours)** | **26.22 / 0.9336** | **27.20 / 0.9018** |
>
> **3. Generalization to unseen distributions**: we evaluate real-world image deraining and remote sensing image deblurring, which come from distributions different from the training data.
>
> **Table: Quantitative generalization comparison on real-world deraining and RSI deblurring.**
>
> | | SPA-Data (Deraining) | UCMerced (RSI Deblurring) |
> |--------|------------------------|----------------------------|
> | Method | PSNR / SSIM | PSNR / SSIM | PSNR / SSIM | PSNR / SSIM |
> | **Single-Task Method** |||
> | MPRNet | 31.99 / 0.9263 | 26.70 / 0.7741 |
> | Restormer | **32.35 / 0.9282** | 27.21 / 0.7928 |
> | NAFNet | 31.99 / 0.9249 | 26.92 / 0.7864 |
> | **All-in-One Method** |||
> | Transweather | 32.21 / 0.9304 | 26.60 / 0.7684 |
> | PromptIR | 32.09 / 0.9254 | 26.77 / 0.7828 |
> | AdaIR | 32.28 / 0.9278 | 27.22 / 0.7951 |
> | MoCEIR | 32.20 / 0.9261 | 27.35 / 0.7956 |
> | **Multi-Domain All-in-One Method** |||
> | **DATPRL-IR (Ours)** | 32.32 / 0.9265 | **28.91 / 0.8108** |
>
> Across all settings, our DATPRL-IR achieves the best or clearly superior performance compared to existing AiOIR methods, demonstrating strong cross-domain, cross-distribution, and cross-task zero-shot and generalization capabilities. These experimental results and detailed analyses are included in the revised version (please see Appendix D.2 for more details).
>
> > Minor typo: The method is referred to as “DATRL-IR” in one sentence of the abstract, missing the “P”.
>
> We thank the reviewer for the careful reading. This was a typo. We have corrected all occurrences to “DATPRL-IR” in the revised version and carefully proofread the manuscript to avoid similar issues.

---

> ### Author Response · Authors · 2025-11-27
>
> > How sensitive is performance to the number of prompts (N=15) and top-k values (k=3/5)? Was this tuned extensively?
>
> Sorry about not being clear on these points. In Table 3, we report the average performance across six tasks under different choices of the prompt number N and the top-k values. The results show two consistent trends. First, increasing the prompt pool to a moderate size improves performance by providing richer options, while an overly large pool yields diminishing or even negative returns, as redundant prompts may dilute useful signals. Second, for top-k selection, using too few prompts limits expressiveness, whereas using too many reduces specificity. A balanced configuration preserves specificity while better leveraging both shared and specific knowledge. For your convenience, we present the results in table below.
>
> **Table: Effect of prompt numbers and top-k selection**
> | **TP Pool** |    **TP Pool**     | **DP Pool** |    **DP Pool**   | **Aver. Performance** |
> |-------------|-------|-------------|-------|------------------------------|
> | Nums | Top-K | Nums | Top-K | PSNR / SSIM |
> | 10 | 1 | 10 | 1 | 30.44 / 0.8607 |
> | 10 | 3 | 10 | 5 | 30.53 / 0.8612 |
> | 15 | 1 | 15 | 1 | 30.48 / 0.8607 |
> | **15** | **3** | **15** | **5** | **30.77 / 0.8653** |
> | 20 | 3 | 20 | 5 | 30.73 / 0.8635 |
> | 20 | 5 | 20 | 5 | 30.70 / 0.8638 |

---

> > ### Comment · Reviewer_mrLL · 2025-11-28
> >
> > Thanks for your detailed reply! I think most of my concerns are addressed.

---

### Official Review · Reviewer_8geW · 2025-10-29

**Soundness:** 3
**Presentation:** 3
**Contribution:** 3
**Rating:** 6
**Confidence:** 3

**Summary:**

The authors propose a novel method called DATPRL-IR, the first model for Multi-Domain All-in-One Image Restoration (MD-AiOIR), which aims to handle diverse restoration tasks across multiple image domains within a single network. Specifically, The author introduces Domain-Aware Task Prompt Representation Learning, which utilizes separate task and domain prompt pools to adaptively retrieve and combine knowledge, subsequently fusing them into a unified guidance signal for the restoration backbone. Extensive experiments demonstrate that DATPRL-IR significantly outperforms existing state-of-the-art methods on various tasks across natural, medical, and remote sensing image domains, showing strong performance and scalability as tasks are added.

**Strengths:**

The article is well-structured and easy to understand. By separating image types and restoration tasks, the author introduces an interesting approach to achieve different types of image restoration. This method avoids the confusion between types and tasks, allowing the model to better perform specific tasks in specific domains.

**Weaknesses:**

1. Motivation: The authors propose a unified image restoration model capable of handling various types of images. This raises several questions: Why is this approach necessary, and why were previous methods unable to achieve this? The authors briefly mention in the introduction that earlier methods did not accomplish such unification, but the author fails to analyze why those methods were incapable of doing so.

2. How to hand multi-modal data inputs: The paper addresses multiple domains but does not explicitly detail how the model handles the significant variations in input data characteristics across these domains. For instance, MRI data is often complex-valued and may be stored as 2-channel data (real and imaginary parts), while natural images are typically 3-channel RGB data. How does the proposed method unify these different data types and channel dimensions to form a consistent input representation for the single backbone network? A clarification of this pre-processing or adaptation step is crucial for understanding the model's practical applicability and reproducibility.

**Questions:**

1. Fusion of domain and task prompts: The direct cross-attention between domain prompt features (PR_d) and task prompt features (PR_t) might not be the most effective fusion strategy. Given that domain and task information are conceptually orthogonal (one specifies the "where," the other the "what"), directly fusing them could lead to information loss or unwanted interference. Would a sequential fusion strategy be more appropriate? For example, the domain prompt could first interact with the image features via cross-attention to produce domain-aware image features. These enriched features could then interact with the task prompts. This hierarchical approach might better preserve the distinct nature of both information types.

2. The design of prompts: I would like to discuss the design of the prompt with the author:
Domain Prompt Pool: The current strategy of relying on adjectives for domain identification appears potentially ambiguous. For instance, the descriptor "black and white" could apply to an old natural image or a medical image. This ambiguity might lead to inaccuracies in domain recognition. Have the authors considered using more explicit, categorical domain names as prompts, such as "This is an MRI image"? Such direct identifiers could provide a more robust and unambiguous signal for domain-aware guidance.
Task Prompt Pool: The manuscript does not sufficiently elaborate on the concrete form or semantics of the task prompts. It remains unclear what kind of information they encapsulate (e.g., degradation patterns, desired output properties). To enhance clarity, could the authors provide a concrete example of what a task prompt might represent, either in the text or by annotating Figure 2? This would help readers better distinguish between the roles of task prompts and domain prompts.

---

> ### Author Response · Authors · 2025-11-27
>
> > Motivation: The authors propose a unified image restoration model capable of handling various types of images. This raises several questions: Why is this approach necessary, and why were previous methods unable to achieve this? [...]
>
> We sincerely thank the reviewer for these insightful questions.
>
> To the best of our knowledge, AiOIR across multiple domains has not been explored previously. As discussed in the Introduction and in Section 3.1 (Motivation), most existing AiOIR methods operate within a single domain (e.g., natural scenes, medical imaging, or remote sensing) [1–6]. These methods do not incorporate domain priors or model relationships across domains—factors that are crucial for MD-AiOIR, as supported by our results in Table 1 and Section 4. For example, several representative AiOIR models, including PromptIR, AdaIR, DFPIR, and AMIR, perform even worse than the single-task baseline Restormer under the multi-domain setting.
>
> Our work specifically targets the MD-AiOIR problem and proposes modeling domain priors to enable AiOIR across heterogeneous domains. Experiments show that our approach consistently outperforms existing AiOIR baselines under this more challenging scenario, underscoring the need to rethink and redesign restoration models for MD-AiOIR. We have clarified this point more explicitly in the revised version and hope this resolves the misunderstanding.
>
> > How to hand multi-modal data inputs: The paper addresses multiple domains but does not explicitly detail how the model handles the significant variations in input data characteristics across these domains. [...]
>
> We apologize for the lack of clarity. In our implementation, all inputs are converted into a standard 3-channel image before being fed into the model. Natural and remote sensing datasets are already in RGB format. For medical datasets that are not RGB, we first convert them to grayscale and then replicate the single channel three times to obtain a 3-channel representation. All images are stored in a consistent format (PNG or JPG), ensuring uniform dimensionality. This clarification has been added to the revised version. (Please see Appendix C.)
>
> > Fusion of domain and task prompts: The direct cross-attention between domain prompt features and task prompt features might not be the most effective fusion strategy. [...]
>
> We thank the reviewer for this insightful suggestion. We agree that sequential fusion is a reasonable alternative and have implemented a “domain → image → task” variant for comparison. As shown in the table below, while sequential fusion achieves strong performance, it still performs slightly worse than our direct fusion. This may be because, in MD-AiOIR, both domain and task prompt representations are equally important. Direct fusion allows the model to learn a prompt representation that is simultaneously task-aware and domain-aware, providing stronger guidance to the restoration network. More details are provided in Appendix E of the revised version.
>
> **Effectiveness of Different Fusion Strategies on 3 Domains & 9 Tasks. Best results in **bold**.**
>
> | | Deraining (Rain100L) | Deblurring (GoPro) | CT Denoising (AAPM) | RSI SR (UCMerced) | Cloud Removal (CUHK) | RSI Dehazing (RICE1) |
> |--------|------------------------|----------------------|------------------------|----------------------|-----------------------------|-------------------------|
> | Method | PSNR / SSIM | PSNR / SSIM | PSNR / SSIM | PSNR / SSIM |  PSNR / SSIM | PSNR / SSIM |
> | Sequential Fusion  | 39.60 / 0.9865 | 29.30 / 0.8832 | 33.76 / 0.9271 | 28.27 / 0.7902 | 25.82 / 0.7577 | 26.76 / 0.9376 |
> | Ours | **39.67 / 0.9867** | **29.57 / 0.8881** | **33.77 / 0.9273** | **28.31 / 0.7913** | **26.00 / 0.7592** | **26.94 / 0.9347** |

---

> ### Author Response · Authors · 2025-11-27
>
> > The design of prompts: I would like to discuss the design of the prompt with the author: Domain Prompt Pool: The current strategy of relying on adjectives for domain identification appears potentially ambiguous.[...]
>
> Many thanks for these valuable suggestions.
>
> **1.** While some adjectives can appear in multiple domains, our design does not rely on text descriptions directly. Instead, we learn a domain prompt pool, which allows the model to perform instance-adaptive selection rather than applying a fixed domain label to all images. This enables the model to leverage shared knowledge across domains while differentiating domain-specific cues. As suggested, we conducted experiments replacing the domain prompt pool with fixed text descriptions (e.g., “This is an MRI image”, “This is a natural image”, etc.). In addition, as raised by Reviewer mrLL, we also evaluated simpler domain encodings (one learnable tensor per domain). Results in the table below show that both alternatives underperform the domain prompt pool across all tasks. Moreover, using explicit prompts requires the user to know the domain beforehand, which conflicts with our blind restoration setting.
>
> **Table: Effectiveness of Different Prompt Designs on 3 Domains & 9 Tasks. Best results are in **bold**.**
>
> | Method | Deraining (Rain100L) | Deblurring (GoPro) | MRI SR (IXI MRI) | PET Synthesis (PolarStar) | Cloud Removal (CUHK CR1) | RSI Dehazing (RICE1) |
> |--------|------------------------|----------------------|---------------------|-----------------------------|-----------------------------|-------------------------|
> | Explicit Domain Prompts | 39.59 / 0.9865 | 29.31 / 0.8839 | 27.77 / 0.9035 | 37.08 / 0.9500 | 25.76 / 0.7573 | 25.91 / 0.9296 |
> | Simple Domain Encodings | 39.52 / 0.9863 | 29.21 / 0.8814 | 27.70 / 0.9031 | 37.02 / 0.9496 | 25.81 / 0.7566 | 25.81 / 0.9213 |
> | **Domain Prompt Pool (Ours)** | **39.67 / 0.9867** | **29.57 / 0.8881** | **27.86 / 0.9045** | **37.12 / 0.9502** | **26.00 / 0.7592** | **26.94 / 0.9347** |
>
> **2.** Task prompts are learned vectors and cannot be visualized directly. However, multiple empirical analyses demonstrate that they capture meaningful information. As shown in Figure 6, visualizations of task prompt selection distribution, prompt diversity, and instance-level prompt representations show that:
> - Each prompt captures distinct content with little redundancy, providing diverse options for the model.
> - Selection distributions of the task prompt pool overlap across datasets, suggesting effective leveraging of shared knowledge across tasks.
> - For the same task, prompt representations have similar overall directions while retaining instance-specific variations.

---

### Official Review · Reviewer_1CdE · 2025-10-31

**Soundness:** 2
**Presentation:** 2
**Contribution:** 2
**Rating:** 2
**Confidence:** 4

**Summary:**

This paper proposes DATPRL-IR, a new method for image restoration across multiple domains (natural, medical, and remote sensing) and tasks, under the umbrella of what the authors term "Multi-domain All-in-One Image Restoration" (MD-AiOIR). The core idea is to utilize a dual prompt pool mechanism — task prompts and domain prompts — and fuse them via a prompt composition mechanism and cross-attention to guide the restoration process. The authors argue that DATPRL-IR is the first to tackle MD-AiOIR, claiming it represents a new research paradigm. Extensive experiments are conducted across 6 and 9 tasks on datasets from different domains.

**Strengths:**

1. The implementation is comprehensive, involving a structured dual-prompt architecture with cross-modal alignment using LLMs (LLaVA/CLIP), regularization mechanisms, and adaptive fusion.
2. Experiments are conducted on a relatively large number of tasks and datasets, covering natural, remote sensing, and medical images.
3. Ablation studies and visualizations are detailed and well-presented, providing insights into the behavior of the model.

**Weaknesses:**

1. The central claim that "Multi-domain All-in-One Image Restoration" is a novel and meaningful research problem is not sufficiently justified. The distinction between MD-AiOIR and regular AiOIR is weak. The proposed setting is essentially a standard multi-task image restoration framework with more diverse datasets included. The fact that earlier works did not include medical or remote sensing datasets does not inherently make this a new problem. The paper does not provide a convincing motivation or theoretical/practical evidence that justifies treating MD-AiOIR as a distinct research direction.
2. The paper fails to clarify whether the compared methods (e.g., PromptIR, MoCE-IR) were re-trained on the same multi-domain setting or simply evaluated out-of-domain. This is critical because these works were not originally designed for cross-domain tasks. Without a fair comparison, i.e., re-training all baselines on the same union of datasets, the reported superiority of DATPRL-IR is not convincing.
3. The idea of using prompts (explicit or implicit) to guide image restoration is not new. Prior works such as PromptIR and related prompt-based multi-task models have explored similar mechanisms. While the dual-prompt pool design sounds well-structured, it is not fundamentally innovative. The use of CLIP or LLaVA to generate text features for domain alignment is incremental and does not offer deep conceptual novelty.
4. Although the paper presents a complex architecture, it does not offer new insights into the nature of task/domain relationships in image restoration. There is little discussion on what has been learned, what the limitations are, or what implications this has for future research in AiOIR or multi-task learning more broadly.

**Questions:**

1. Were the compared methods (e.g., PromptIR, MoCE-IR) retrained on the same combined dataset of natural, medical, and remote sensing images? If not, how can we interpret the performance gap?
2. How do the authors justify the claim that MD-AiOIR is a new research paradigm, rather than an empirical extension of existing AiOIR to more datasets?
3. Given that medical and remote sensing datasets are relatively small in this paper, could the performance gains be due to overfitting or data imbalance?
4. What are the broader implications of this work for the field of image restoration? Does the method generalize to unseen domains or tasks?

---

> ### Author Response · Authors · 2025-11-27
>
> > The central claim that “Multi-domain All-in-One Image Restoration” is a novel and meaningful research problem is not sufficiently justified. The distinction between MD-AiOIR and regular AiOIR is weak. The proposed setting is essentially a standard multi-task image restoration framework with more diverse datasets included. [...]
>
> Thanks for raising this important point. We apologize for not making it sufficiently clear. MD-AiOIR is not simply “AiOIR with more datasets”.
>
> To the best of our knowledge, AiOIR across multiple domains has not been explored previously. As discussed in the Introduction and in Section 3.1 (Motivation), most existing AiOIR methods operate within a single domain (e.g., natural scenes, medical imaging, or remote sensing) [1–6]. These methods do not incorporate domain priors or model relationships across domains—factors that are crucial for MD-AiOIR, as supported by our results in Table 1 and Section 4. For example, several representative AiOIR models, including PromptIR, AdaIR, DFPIR, and AMIR, perform even worse than the single-task baseline Restormer under the multi-domain setting.
>
> Our work specifically targets the MD-AiOIR problem and proposes modeling domain priors to enable AiOIR across heterogeneous domains. Experiments show that our approach consistently outperforms existing AiOIR baselines under this more challenging scenario, underscoring the need to rethink and redesign restoration models for MD-AiOIR. We have clarified this point more explicitly in the revised version and hope this resolves the misunderstanding.
>
> [1] Zhiwen Yang, Haowei Chen, Ziniu Qian, Yang Yi, Hui Zhang, Dan Zhao, Bingzheng Wei, and Yan Xu. All-in-one medical image restoration via task-adaptive routing. MICCAI, 2024.
>
> [2] Zhiwen Yang, Jiayin Li, Hui Zhang, Dan Zhao, Bingzheng Wei, and Yan Xu. Restore-rwkv: Efficient and effective medical image restoration with rwkv. IEEE J-BHI, 2025.
>
> [3] Haowei Chen, Zhiwen Yang, Haotian Hou, Hui Zhang, Bingzheng Wei, Gang Zhou, and Yan Xu. All-in-one medical image restoration with latent diffusion-enhanced vector-quantized codebook prior. MICCAI, 2025.
>
> [4] Vaishnav Potlapalli, Syed Waqas Zamir, Salman H Khan, and Fahad Shahbaz Khan. Promptir: Prompting for all-in-one image restoration. NeurIPS, 2023.
>
> [5] Xiangpeng Tian, Xiangyu Liao, Xiao Liu, Meng Li, and Chao Ren. Degradation-aware feature perturbation for all-in-one image restoration. CVPR, 2025.
>
> [6] Boyun Li, Xiao Liu, Peng Hu, Zhongqin Wu, Jiancheng Lv, and Xi Peng. All-in-one image restoration for unknown corruption. CVPR, 2022.
>
> > The paper fails to clarify whether the compared methods (e.g., PromptIR, MoCE-IR) were re-trained on the same multi-domain setting or simply evaluated out-of-domain. [...]
>
> We appreciate the question. As stated in Section 4.1 (Implementation Details), all competing methods were trained from scratch on the same combined datasets as ours, using their original training strategies as described in their papers. To ensure a fully fair comparison, all remaining training conditions were kept identical to ours, including the train/test split, domain-balanced sampling, data augmentations, and other shared hyperparameters. We have made this point clearer in the revised version to avoid any ambiguity. (Please see Appendix C)
>
> > The idea of using prompts (explicit or implicit) to guide image restoration is not new. Prior works such as PromptIR [...]. While the dual-prompt pool design sounds well-structured, [...]
>
> Thanks for the comment. With full respect, we did not claim that using prompts for image restoration is our contribution. Our methodological contribution lies in using the dual-prompt-pool design to leverage both shared and specific knowledge across diverse domains and tasks for MD-AiOIR. In contrast, existing prompt-based AiOIR methods (e.g., PromptIR, DFPIR) focus primarily on task differences [1, 2] and adopt prompts to maximize task separability, which becomes problematic as domains and tasks scale. These works were properly cited and positioned in our paper.
>
> [1] Vaishnav Potlapalli, Syed Waqas Zamir, Salman H Khan, and Fahad Shahbaz Khan. Promptir: Prompting for all-in-one image restoration. NeurIPS, 2023.
>
> [2] Xiangpeng Tian, Xiangyu Liao, Xiao Liu, Meng Li, and Chao Ren. Degradation-aware feature perturbation for all-in-one image restoration. CVPR, 2025.

---

> ### Author Response · Authors · 2025-11-27
>
> > Although the paper presents a complex architecture, it does not offer new insights into the nature of task/domain relationships in image restoration. [...]
>
> Many thanks for the constructive suggestion. We have added a dedicated discussion section in the revised version (please see Appendix B).
>
> In this paper, we show that different domains/tasks share meaningful common knowledge while still retaining specific characteristics (as stated in "Introduction" and "Motivation in Section 3.1"). Modeling this “shared + specific” structure reduces learning difficulty as the number of domains and tasks grows, and improves restoration performance. We believe this provides a new perspective for building more unified and scalable restoration models. We also acknowledge limitations: our training cost is higher than that of single-domain or single-task approaches, and while our method shows strong zero-shot behavior, extending it to a wider range of domains remains promising future work. Further exploration of how to model shared and distinctive priors more efficiently and interpretably is also an important direction.
>
> > Given that medical and remote sensing datasets are relatively small in this paper, could the performance gains be due to overfitting or data imbalance?
>
> Thanks a lot for the question.
>
> - Although dataset sizes vary, we mitigate imbalance through extensive data augmentation (random cropping, flipping, rotation) and, more importantly, a task-balanced dataloader that ensures each batch contains the same number of samples from every task. All competing methods are trained under exactly the same strategy, ensuring a fair comparison.
>
> - In addition, our zero-shot and generalization experiments show strong performance on unseen domains and tasks, suggesting that the model is not overfitting to specific datasets.
>
> Therefore, the performance gains are not attributable to overfitting or data imbalance.

---

> ### Author Response · Authors · 2025-11-27
>
> > What are the broader implications of this work for the field of image restoration? Does the method generalize to unseen domains or tasks?
>
> We appreciate the reviewer’s questions.
>
> This work takes the first step toward multi-domain all-in-one image restoration, pushing the field toward more unified restoration frameworks. Our study highlights that domains and tasks contain both shared and specific priors, and leveraging this structure reduces learning difficulty in large, heterogeneous settings. We believe this offers a new perspective for unified image restoration.
>
> Regarding generalization, we have added three types of new evaluations in the revised version. All results are obtained using the well-trained models without any fine-tuning. Please refer to Tables 6–8 and Figures 10–13 in the Appendix for detailed quantitative results and visual comparisons.
>
> **1. Zero-shot on unseen domains**: we test performance on two domains not seen during training: AIGC images and comic images.
>
> **Table: Quantitative zero-shot SR comparison on AIGC and Comic domains. Best metrics are in bold.**
>
> | | SDXL-Turbo | DeepFloyd | Midjourney | Manga109 |
> |--------|------------|-----------|------------|----------|
> | Method | PSNR / SSIM | PSNR / SSIM | PSNR / SSIM | PSNR / SSIM |
> | **Single-Task Method** |||||
> | MPRNet | 31.13 / 0.8730 | 26.76 / 0.8254 | 27.19 / 0.7656 | 27.13 / 0.8612 |
> | SwinIR | 31.20 / 0.8709 | 27.41 / 0.8344 | 27.19 / 0.7627 | 27.76 / 0.8697 |
> | Restormer | 32.95 / 0.8985 | 28.81 / 0.8675 | 28.24 / 0.8005 | 28.13 / 0.8758 |
> | NAFNet | 32.67 / 0.8959 | 28.59 / 0.8637 | 28.12 / 0.7965 | 27.98 / 0.8727 |
> | **All-in-One Method** |||||
> | Transweather | 32.49 / 0.8916 | 27.73 / 0.8456 | 27.81 / 0.7866 | 26.00 / 0.8330 |
> | PromptIR | 32.38 / 0.8933 | 28.56 / 0.8642 | 27.90 / 0.7916 | 28.28 / 0.8756 |
> | AdaIR | 32.95 / 0.8989 | 28.87 / 0.8687 | 28.23 / 0.8008 | 28.05 / 0.8758 |
> | MoCEIR | 33.00 / 0.8995 | 28.90 / 0.8680 | 28.30 / 0.8027 | 28.10 / 0.8745 |
> | **Multi-Domain All-in-One Method** |||||
> | **DATPRL-IR (Ours)** | **33.25 / 0.9017** | **29.07 / 0.8713** | **28.38 / 0.8045** | **28.73 / 0.8828** |
>
> **2. Zero-shot on completely unseen tasks**: we further test the models on two tasks not included in training: natural image desnowing and specular highlight removal.
>
> **Table: Quantitative zero-shot comparison on specular highlight removal and desnowing  .**
>
> |  | SHIQ (Highlight Removal) | Snow100K-S (Desnowing) |
> |--------|----------------------------|--------------------------|
> | Method | PSNR / SSIM | PSNR / SSIM | PSNR / SSIM | PSNR / SSIM |
> | **Single-Task Method** |||
> | MPRNet | 25.83 / 0.9297 | 26.09 / 0.8882 |
> | Restormer | 25.67 / 0.9295 | 26.57 / 0.8955 |
> | NAFNet | 25.89 / 0.9298 | 26.27 / 0.8923 |
> | **All-in-One Method** |||
> | Transweather | 26.20 / 0.9315 | 26.36 / 0.8870 |
> | PromptIR | 24.87 / 0.9239 | 26.42 / 0.8923 |
> | AdaIR | 25.35 / 0.9263 | 26.54 / 0.8951 |
> | MoCEIR | 25.30 / 0.9259 | 26.65 / 0.8976 |
> | **Multi-Domain All-in-One Method** |||
> | **DATPRL-IR (Ours)** | **26.22 / 0.9336** | **27.20 / 0.9018** |
>
> **3. Generalization to unseen distributions**: we evaluate real-world image deraining and remote sensing image deblurring, which come from distributions different from the training data.
>
> **Table: Quantitative generalization comparison on real-world deraining and RSI deblurring.**
>
> | | SPA-Data (Deraining) | UCMerced (RSI Deblurring) |
> |--------|------------------------|----------------------------|
> | Method | PSNR / SSIM | PSNR / SSIM | PSNR / SSIM | PSNR / SSIM |
> | **Single-Task Method** |||
> | MPRNet | 31.99 / 0.9263 | 26.70 / 0.7741 |
> | Restormer | **32.35 / 0.9282** | 27.21 / 0.7928 |
> | NAFNet | 31.99 / 0.9249 | 26.92 / 0.7864 |
> | **All-in-One Method** |||
> | Transweather | 32.21 / 0.9304 | 26.60 / 0.7684 |
> | PromptIR | 32.09 / 0.9254 | 26.77 / 0.7828 |
> | AdaIR | 32.28 / 0.9278 | 27.22 / 0.7951 |
> | MoCEIR | 32.20 / 0.9261 | 27.35 / 0.7956 |
> | **Multi-Domain All-in-One Method** |||
> | **DATPRL-IR (Ours)** | 32.32 / 0.9265 | **28.91 / 0.8108** |
>
> Across all settings, our DATPRL-IR achieves the best or clearly superior performance compared to existing AiOIR methods, demonstrating strong cross-domain, cross-distribution, and cross-task zero-shot and generalization capabilities. Our learned prompt pools capture reusable domain and task priors rather than dataset-specific patterns. When tested on unseen domains, tasks, or distributions, our model can adapt by selecting and composing prompts that best match the new data distribution. These experimental results and detailed analyses are included in the revised version (please see Appendix D.2 for more details).

---

### Official Review · Reviewer_5gy3 · 2025-10-31

**Soundness:** 3
**Presentation:** 3
**Contribution:** 3
**Rating:** 4
**Confidence:** 4

**Summary:**

The paper proposes DATPRL-IR, a multi-domain all-in-one image restoration framework that integrates both task-specific and domain-specific priors via a dual prompt-pool mechanism. It adaptively selects task and domain prompts, aligns domain prompts with textual features from LLaVA/CLIP, and fuses them through cross-attention to guide restoration. Experiments on 6- and 9-task, 3-domain setups show clear gains over SOTA AiOIR methods in PSNR/SSIM and generalization ability.

**Strengths:**

1. The problem scope is novel and interesting, from my point of view, this is the first attempt to unify AiOIR across multiple task domains.

2. The dual prompt pools and cross-modal alignment are practically effective, and supported by comprehensive experiments with strong empirical performance and ablation studies.

3. Figures and paper writings are easy to follow.

**Weaknesses:**

1. Although the evaluation scope contains multiple tasks from three domains, it is still unclear how the method will perform under zero-shot unseen domains. I think this is important to provide related experimental results to evaluate whether the proposed method is truely practical across domains.
2. LLaVa seems generate captions that may exceed the 77 tokens limit of CLIP text encoder. Which part of the text is useful for cross modal alignment is worth exploration.
3. The authors only provide PSNR/SSIM results for Table 1, lacking subjective metrics such as MANIQA/CLIPIQA etc., making it hard to fully validate the effectiveness of the proposed method.
4. In abstract, the method is called "DATRL-IR" and "DATPRL-IR", which may be a typo.

**Questions:**

For now I prefer a 4 rating (because no 5 option), but I will raise score if the weakness and the questions are properly addressed.

1. How sensitive is the method to the choice of MLLM (e.g., LLaVA vs smaller models)?
2. Are domain prompts reusable or transferable to unseen domains?

---

> ### Author Response · Authors · 2025-11-27
>
> > Although the evaluation scope contains multiple tasks from three domains, it is still unclear how the method will perform under zero-shot unseen domains. [...]
>
> Many thanks for the suggestion. To more comprehensively demonstrate the zero-shot and generalization capabilities of our model, we have added three new types of evaluations in the revised version (Please see Appendix D.2). All experiments use the well-trained models with no fine-tuning applied. Please refer to Tables 6–8 and Figures 10–13 in the Appendix for detailed quantitative results and visual comparisons.
>
> **1. Zero-shot on unseen domains**: we test performance on two domains not seen during training: AIGC images and comic images.
>
> **Table: Quantitative zero-shot SR comparison on AIGC and Comic domains. Best metrics are in bold.**
>
> | | SDXL-Turbo | DeepFloyd | Midjourney | Manga109 |
> |--------|------------|-----------|------------|----------|
> | Method | PSNR / SSIM | PSNR / SSIM | PSNR / SSIM | PSNR / SSIM |
> | **Single-Task Method** |||||
> | MPRNet | 31.13 / 0.8730 | 26.76 / 0.8254 | 27.19 / 0.7656 | 27.13 / 0.8612 |
> | SwinIR | 31.20 / 0.8709 | 27.41 / 0.8344 | 27.19 / 0.7627 | 27.76 / 0.8697 |
> | Restormer | 32.95 / 0.8985 | 28.81 / 0.8675 | 28.24 / 0.8005 | 28.13 / 0.8758 |
> | NAFNet | 32.67 / 0.8959 | 28.59 / 0.8637 | 28.12 / 0.7965 | 27.98 / 0.8727 |
> | **All-in-One Method** |||||
> | Transweather | 32.49 / 0.8916 | 27.73 / 0.8456 | 27.81 / 0.7866 | 26.00 / 0.8330 |
> | PromptIR | 32.38 / 0.8933 | 28.56 / 0.8642 | 27.90 / 0.7916 | 28.28 / 0.8756 |
> | AdaIR | 32.95 / 0.8989 | 28.87 / 0.8687 | 28.23 / 0.8008 | 28.05 / 0.8758 |
> | MoCEIR | 33.00 / 0.8995 | 28.90 / 0.8680 | 28.30 / 0.8027 | 28.10 / 0.8745 |
> | **Multi-Domain All-in-One Method** |||||
> | **DATPRL-IR (Ours)** | **33.25 / 0.9017** | **29.07 / 0.8713** | **28.38 / 0.8045** | **28.73 / 0.8828** |
>
> **2. Zero-shot on completely unseen tasks**: we further test the models on two tasks not included in training: natural image desnowing and specular highlight removal.
>
> **Table: Quantitative zero-shot comparison on specular highlight removal and desnowing  .**
>
> |  | SHIQ (Highlight Removal) | Snow100K-S (Desnowing) |
> |--------|----------------------------|--------------------------|
> | Method | PSNR / SSIM | PSNR / SSIM | PSNR / SSIM | PSNR / SSIM |
> | **Single-Task Method** |||
> | MPRNet | 25.83 / 0.9297 | 26.09 / 0.8882 |
> | Restormer | 25.67 / 0.9295 | 26.57 / 0.8955 |
> | NAFNet | 25.89 / 0.9298 | 26.27 / 0.8923 |
> | **All-in-One Method** |||
> | Transweather | 26.20 / 0.9315 | 26.36 / 0.8870 |
> | PromptIR | 24.87 / 0.9239 | 26.42 / 0.8923 |
> | AdaIR | 25.35 / 0.9263 | 26.54 / 0.8951 |
> | MoCEIR | 25.30 / 0.9259 | 26.65 / 0.8976 |
> | **Multi-Domain All-in-One Method** |||
> | **DATPRL-IR (Ours)** | **26.22 / 0.9336** | **27.20 / 0.9018** |
>
> **3. Generalization to unseen distributions**: we evaluate real-world image deraining and remote sensing image deblurring, which come from distributions different from the training data.
>
> **Table: Quantitative generalization comparison on real-world deraining and RSI deblurring.**
>
> | | SPA-Data (Deraining) | UCMerced (RSI Deblurring) |
> |--------|------------------------|----------------------------|
> | Method | PSNR / SSIM | PSNR / SSIM | PSNR / SSIM | PSNR / SSIM |
> | **Single-Task Method** |||
> | MPRNet | 31.99 / 0.9263 | 26.70 / 0.7741 |
> | Restormer | **32.35 / 0.9282** | 27.21 / 0.7928 |
> | NAFNet | 31.99 / 0.9249 | 26.92 / 0.7864 |
> | **All-in-One Method** |||
> | Transweather | 32.21 / 0.9304 | 26.60 / 0.7684 |
> | PromptIR | 32.09 / 0.9254 | 26.77 / 0.7828 |
> | AdaIR | 32.28 / 0.9278 | 27.22 / 0.7951 |
> | MoCEIR | 32.20 / 0.9261 | 27.35 / 0.7956 |
> | **Multi-Domain All-in-One Method** |||
> | **DATPRL-IR (Ours)** | 32.32 / 0.9265 | **28.91 / 0.8108** |
>
> Across all settings, our DATPRL-IR achieves the best or clearly superior performance compared to existing AiOIR methods, demonstrating strong cross-domain, cross-distribution, and cross-task zero-shot and generalization capabilities. These experimental results and detailed analyses are included in the revised version (please see Appendix D.2 for more details).
>
> > LLaVa seems generate captions that may exceed the 77 tokens limit of CLIP text encoder. [...]
>
> Thanks a lot for the comment. In our implementation, we instruct the MLLM to produce concise descriptions, and all texts are processed using ***clip.tokenize(..., truncate=True)***, which automatically truncates long texts to 77 tokens. Our statistical analysis shows that fewer than 1.5% of descriptions are truncated.
>
> Furthermore, our word clouds and t-SNE analysis of CLIP text features (Figure 3 in Section 3.1) demonstrates that the texts not only separate different domains but also capture meaningful cross-domain commonalities. This supports our main motivation—leveraging both specific and shared domain priors to guide restoration. These results indicate that our implementation is sufficient for stable cross-modal alignment and effective domain prompt learning.

---

> ### Author Response · Authors · 2025-11-27
>
> > The authors only provide PSNR/SSIM results for Table 1, lacking subjective metrics such as MANIQA/CLIPIQA etc., [...].
>
> We appreciate the reviewer’s comment. As suggested, we have added five perceptual metrics—CLIPIQA, MANIQA, MUSIQ, NIQE, and FID—under the 3-domain, 9-task setting in the revised version. Across all evaluated tasks, DATPRL-IR achieves consistently better or comparable perceptual scores compared to competing methods. The full results and detailed analyses are provided in Appendix D.3.
>
> **Table: Perceptual Quality Comparison on 3 Domains & 9 Tasks . Best metrics are in **bold**.**
>
> | |  |  | SR |  |  |  |  | Derain  | |  |  |  | Deblur  ||  |
> |--------|----------------|--|--|--|--|------------------------|--|--|--|--|---------------------|--|--|--|--|
> |    Method     | CLIPIQA ↑ | MANIQA ↑ | MUSIQ ↑ | NIQE ↓ | FID ↓ | CLIPIQA ↑ | MANIQA ↑ | MUSIQ ↑ | NIQE ↓ | FID ↓ | CLIPIQA ↑ | MANIQA ↑ | MUSIQ ↑ | NIQE ↓ | FID     ↓ |
> | MPRNet | 0.4561 | 0.4998 | 55.35 | 5.9716 | 104.76 | 0.7480 | 0.6967 | 70.61 | **3.1747** | 9.0289 | 0.3218 | 0.4116 | 34.01 | 5.6452 | 20.73 |
> | SwinIR | 0.4512 | 0.5059 | 58.06 | 5.9690 | 102.31 | 0.7560 | 0.6987 | 70.63 | 3.1858 | 8.3961 | 0.3142 | 0.4068 | 35.91 | 5.6113 | 25.26 |
> | Restormer | 0.4634 | 0.5138 | **58.28** | 6.0288 | 102.01 | 0.7543 | 0.7010 | 70.85 | 3.2066 | 6.5893 | 0.3468 | 0.4429 | 36.96 | 5.5778 | 17.05 |
> | NAFNet | 0.4607 | 0.5041 | 56.10 | **5.9210** | 102.45 | 0.7530 | 0.6976 | 70.68 | 3.2944 | 8.9292 | 0.3410 | 0.4410 | 37.24 | 5.4487 | 16.77 |
> | TransWeather | 0.4500 | 0.4828 | 51.85 | 6.6003 | 102.20 | 0.6803 | 0.6682 | 69.07 | 3.4578 | 33.03 | 0.3023 | 0.3472 | 27.81 | 6.1708 | 36.66 |
> | PromptIR | 0.4585 | 0.5093 | 56.48 | 6.2211 | 101.95 | 0.7545 | 0.6998 | 70.81 | 3.2141 | 7.5467 | 0.3385 | 0.4382 | 36.39 | 5.6221 | 18.25 |
> | AdaIR | 0.4624 | **0.5161** | 58.18 | 6.0680 | 102.18 | 0.7556 | 0.7010 | 70.86 | 3.2149 | 6.8281 | 0.3201 | 0.4547 | **38.07** | 5.5143 | 16.79 |
> | MoCEIR | 0.4604 | 0.5120 | 55.47 | 6.0852 | 102.98 | 0.7573 | 0.7011 | 70.85 | 3.2167 | 7.1322 | 0.3270 | **0.4597** | 37.87 | 5.5175 | 16.88 |
> | **DATPRL-IR (Ours)** | **0.5558** | 0.5156 | 56.29 | 6.2490 | **101.93** | **0.7587** | **0.7033** | **70.94** | 3.2410 | **5.1064** | **0.3485** | 0.4461 | 35.93 | **5.4463** | **16.73** |
>
> We would like to note that we report these perceptual metrics only on natural-image tasks. Metrics such as CLIPIQA, MANIQA, and MUSIQ are pre-trained exclusively on natural image datasets, so applying them to medical or remote sensing images would not yield reliable or meaningful results. Moreover, both the medical/remote sensing communities and the all-in-one restoration literature primarily focus on fidelity-based (pixel-level) metrics [1-8].
>
> [1] Zhiwen Yang, Haowei Chen, Ziniu Qian, Yang Yi, Hui Zhang, Dan Zhao, Bingzheng Wei, and Yan Xu. All-in-one medical image restoration via task-adaptive routing. MICCAI, 2024.
>
> [2] Zhiwen Yang, Jiayin Li, Hui Zhang, Dan Zhao, Bingzheng Wei, and Yan Xu. Restore-rwkv: Efficient and effective medical image restoration with rwkv. IEEE J-BHI, 2025.
>
> [3] Haowei Chen, Zhiwen Yang, Haotian Hou, Hui Zhang, Bingzheng Wei, Gang Zhou, and Yan Xu. All-in-one medical image restoration with latent diffusion-enhanced vector-quantized codebook prior. MICCAI, 2025.
>
> [4] Hanlin Wu, Ning Ni, and Libao Zhang. Lightweight stepless super-resolution of remote sensing images via saliency-aware dynamic routing strategy. IEEE TGRS, 2023.
>
> [5] Yi Liu, Wengen Li, Jihong Guan, Shuigeng Zhou, and Yichao Zhang. Effective cloud removal for remote sensing images by an improved mean-reverting denoising model with elucidated design space. CVPR, 2025.
>
> [6] Vaishnav Potlapalli, Syed Waqas Zamir, Salman H Khan, and Fahad Shahbaz Khan. Promptir: Prompting for all-in-one image restoration. NeurIPS, 2023.
>
> [7] Eduard Zamfir, Zongwei Wu, Nancy Mehta, Yuedong Tan, Danda Pani Paudel, Yulun Zhang, and Radu Timofte. Complexity experts are task-discriminative learners for any image restoration. CVPR, 2025.
>
> [8] Boyun Li, Xiao Liu, Peng Hu, Zhongqin Wu, Jiancheng Lv, and Xi Peng. All-in-one image restoration for unknown corruption. CVPR, 2022.
>
> > In abstract, the method is called “DATRL-IR” and “DATPRL-IR”, which may be a typo.
>
> We thank the reviewer for the careful reading. This was a typo. We have corrected all occurrences to “DATPRL-IR” in the revised version and carefully proofread the manuscript to avoid similar issues.

---

> ### Author Response · Authors · 2025-11-27
>
> > How sensitive is the method to the choice of MLLMs? [...]
>
> We appreciate the constructive comments. Our method is not strongly dependent on a specific MLLM. We experimented with replacing LLaVA-1.5-7B with two other models—LLaVA-1.5-13B and Qwen3-VL-2B-Instruct—and observed stable performance with only marginal differences under these MLLMs of different parameter scales. This is because we only rely on relatively coarse domain-level semantics (e.g., “main image content”, “shooting view”, “brightness”, “color” etc.), which can be provided by most general-purpose MLLMs. We have included these results in the revised manuscript (please see Appendix E). We also provide some examples of generated textual descriptions in Table.5.
>
> **Table: Effectiveness of Different MLLMs on 3 Domains & 9 Tasks. Best results are in **bold**.**
>
> | | Natural SR (DIV2K-Val) | Deraining (Rain100L) | CT Denoising (AAPM) | PET Synthesis (PolarStar) | RSI SR (UCMerced) | RSI Dehazing (RICE1) |
> |--------|--------------------------|------------------------|------------------------|----------------------------|----------------------|-------------------------|
> | Method | PSNR / SSIM | PSNR / SSIM | PSNR / SSIM | PSNR / SSIM | PSNR / SSIM | PSNR / SSIM |
> | **LLaVA1.5-13B** | 29.04 / 0.8180 | **39.71 / 0.9868** | **33.77** / 0.9272 | 37.08 / 0.9500 | **28.31 / 0.7918** | 26.92 / 0.9347 |
> | **LLaVA1.5-7B** | 29.05 / 0.8181 | 39.67 / 0.9867 | **33.77** / **0.9273** | **37.12 / 0.9502** | **28.31** / 0.7913 | **26.94** / 0.9347 |
> | **Qwen3-VL-2B-Instruct** | **29.07 / 0.8182** | 39.69 / **0.9868** | **33.77** / 0.9270 | 37.11 / 0.9501 | **28.31** / 0.7914 | 26.82 / **0.9351** |
>
> > Are domain prompts reusable or transferable to unseen domains?
>
> Yes. As shown in our zero-shot and generalization experiments, the domain prompt pool captures reusable domain priors rather than dataset-specific patterns. When tested on unseen domains or tasks, the model adapts by selecting and composing prompts that best match the new data distribution, demonstrating clear transferability. We have added this discussion and the corresponding results in the revised version (please see Appendix D.2).

---

> > ### Comment · Reviewer_5gy3 · 2025-11-28
> >
> > Thanks for the authors' detailed rebuttal. The newly added experimental results under zero-shot settings further support the effectiveness of the proposed prompt mechanism. Results regarding NR-IQA metrics demonstrate that the proposed method generally outperform others. Further experiments on different MLLM choices show that the proposed method only rely on coarse semantics, which indicates the scalability of the proposed method.
> >
> > My concerns and questions are properly addressed, and I'll raise my score. I now think this paper is worth accepting.

---

> ### Comment · Reviewer_5gy3 · 2025-11-28
> **Remark after rebuttal**
>
> I don't know why I can't edit my reviews now. My final rating for this paper would be 6.

---

### Author Response · Authors · 2025-12-03
**A Summary of Our Interactions with Reviewers**

We sincerely thank all Reviewers, ACs, and SACs for their thoughtful and constructive feedback. We are grateful that the reviewers recognized our work for its novel problem setting, its well-motivated and practically effective dual-prompt-pool design, the comprehensive experiments, and the clarity of the paper. Below is a concise summary of our interactions.

***

`Reviewer 5gy3`

**Main concerns:** (1) zero-shot evidence, (2) truncation of LLaVA captions, (3) missing subjective metrics, (4) robustness across MLLMs.

**Our response:**

(1) We added comprehensive zero-shot and generalization evaluations on unseen domains, unseen distributions, and entirely unseen tasks (Appendix D.2).

(2) We provided full implementation details, statistical analysis of CLIP token truncation, and a t-SNE visualization confirming stable semantics (Figure 3, Section 3.1).

(3) We included five perceptual metrics showing clear perceptual advantages (Appendix D.3).

(4) We added comparisons across three MLLMs—LLaVA-1.5-7B/13B and Qwen3-VL-2B—showing stable performance with only marginal differences (Appendix E).

**Reviewer’s follow-up:** “[…] My concerns and questions are properly addressed, and I’ll raise my score. **I now think this paper is worth accepting**. […]”

***

`Reviewer 1CdE`

**Main concerns:** (1) novelty of MD-AiOIR, (2) fairness of baseline retraining, (3) novelty of using prompts, (4) broader insights, (5) data imbalance or overfitting.

**Our response:**

(1) We clarified that existing AiOIR methods do not model domain priors and degrade when scaling to multiple domains, making MD-AiOIR a distinct and previously unexplored setting.

(2) We confirmed that all baselines were retrained fairly with identical datasets, sampling strategies, and augmentations (Appendix C).

(3) We emphasized that our contribution lies not in “using prompts” but in a dual-prompt-pool framework capturing both shared and specific domain/task priors.

(4) We added a summary of broader insights and future directions (Appendix B).

(5) We provided evidence that the improvements do not result from data imbalance or overfitting, supported by strong zero-shot and generalization results (Appendix D.2).

***

`Reviewer 8geW`

**Main concerns:** (1) motivation for a unified multi-domain model, (2) input unification, (3) prompt-fusion strategy, (4) domain-prompt design, (5) task-prompt semantics.

**Our response:**

(1) We expanded the motivation, explaining why prior AiOIR models fail under MD-AiOIR.

(2) We provided full input pre-processing details (Appendix C).

(3) We added an ablation comparing sequential fusion with our direct fusion approach (Appendix E).

(4) We included domain-prompt design ablations using explicit text prompts and simple encodings (Appendix E).

(5) We added visual analyses of task-prompt selection distributions, prompt diversity, and instance-level representations, showing that task prompts capture meaningful semantics (Figure 6, Section 4.4).

***

`Reviewer mrLL`

**Main concerns:** (1) implementation details of task prompts, (2) simpler domain encodings, (3) MLLM caption reliability, (4) zero-shot evidence, (5) sensitivity to the number of prompts and top-k.

**Our response:**

(1) We added detailed explanations of prompt initialization, projector design, and integration.

(2) We included two alternative designs (explicit prompts, simple encodings) demonstrating our method’s advantages (Appendix E).

(3) We added LLaVA-generated textual examples for all domains (Table 5, Appendix C).

(4) We conducted extensive zero-shot and generalization evaluations (Appendix D.2).

(5) We added thorough ablations on the number of prompts and top-k values (Table 3, Section 4.3).

**Reviewer’s follow-up:** “Thanks for your detailed reply! I think most of my concerns are addressed.”

***

**All updates are highlighted in blue in the revised manuscript for easy reference.**

---

### Meta-Review · Area_Chair_WAHy · 2025-12-09

**Summary:**

Need zero-shot / generalization evidence to unseen domains, tasks (5gy3, 1CdE, mrLL)

Clarify novelty of the MD-AiOIR setting vs. standard AiOIR; is the dual-prompt idea truly new? (1CdE)

Ensure fair baselines (retraining on the same multi-domain mix with equal protocols) (1CdE)

Risk of overfitting / data imbalance (small medical/RS sets) (1CdE)

Motivation for a unified multi-domain model; why prior AiOIR couldn’t do this (8geW)

How inputs from very different domains are unified/preprocessed (e.g., grayscale MRI, complex-valued data) (8geW)

Whether the chosen prompt-fusion strategy (direct cross-attention) is optimal vs sequential/hierarchical fusion (8geW)

Domain-prompt design may be ambiguous (adjectives); consider explicit domain labels; clarify task-prompt semantics (8geW)

Missing details on task-prompt learning (key/value coupling, projector, how values condition the backbone) (mrLL)

Magnitude/necessity of domain-prompt gains; would simpler encodings (one-hot/embeddings) suffice? (mrLL)

MLLM robustness: sensitivity to LLaVA vs other MLLMs; CLIP 77-token truncation impact (5gy3)

Lack of subjective/perceptual metrics (e.g., CLIPIQA, MANIQA, MUSIQ, NIQE, FID) (5gy3)

**Reviewer Concerns:**

Fully addressed:

Added extensive zero-shot & generalization results to unseen domains (AIGC/comics), unseen tasks (desnowing, highlight removal), and shifted distributions (real derain, RSI deblur) (5gy3, mrLL)  Both reviewers said their concerns were addressed; 5gy3 raised to 6.

Perceptual metrics added (CLIPIQA, MANIQA, MUSIQ, NIQE, FID) with consistent improvements (5gy3)

MLLM robustness: results across LLaVA-1.5-7B/13B and Qwen3-VL-2B showing marginal differences; plus token-truncation stats (<1.5% truncated) and t-SNE/word-cloud analyses (5gy3)

Detailed task-prompt mechanism (random init, keys/values learned independently, 3-layer CNN projector to 1024-d, fusion via cross-attention and AGF into decoder blocks) (mrLL)

Prompt-count / top-k sensitivity ablation table; guidance on sweet spots (mrLL)


Addressed without reviewer follow-up; I assume to the reviewers' satisfaction:

Baseline fairness: authors state all baselines retrained from scratch on the same combined datasets with identical sampling, splits, augs, and shared hparams; clarified in Section 4.1 & Appendix C (1CdE)

Novelty & motivation of MD-AiOIR: articulated why single-domain AiOIR degrades in multi-domain settings; dual-prompt pools model shared + specific priors; added discussion of broader insights/implications (Appendix B) (1CdE, 8geW)

Overfitting / imbalance: task-balanced dataloader + strong zero-shot results argue against overfitting (1CdE)

Input unification across domains: convert to 3-channel images (grayscale replicated to RGB); clarified storage format (8geW)

Fusion strategy: implemented sequential (domain→image→task) variant; direct fusion still slightly better across tasks (8geW)

Domain-prompt design vs simpler alternatives: compared explicit domain text (“This is an MRI image”) and simple domain encodings (one learnable tensor) — both underperform the learned domain prompt pool (8geW, mrLL)

Task-prompt semantics visibility: provided analyses of selection distributions, diversity, and instance-level representations indicating meaningful, low-redundancy prompts (8geW)

LLaVA description reliability: provided multi-domain caption examples; t-SNE shows clustered yet shared semantics; dependence only on coarse cues (mrLL, also relevant to 5gy3)

Partially addressed:

Is MD-AiOIR a distinct research paradigm? Authors give motivation, empirical evidence, and insights; final acceptance of “paradigm” framing remains a judgment call (1CdE)

**Reviewer Scores:**

5gy3 → 6 (marginally above acceptance) — explicitly said “My final rating… 6.”

1CdE → 2 (reject) — no update after rebuttal shown.

8geW → 6 (marginally above acceptance) — no change recorded post-rebuttal.

mrLL → 6 (marginally above acceptance) — said most concerns addressed; no score change recorded.

So, a final score set of 2, 6, 6, 6 would suggest acceptance. The authors also have shown clear responsiveness to the reviewers' concerns.

---

### Decision · Program_Chairs · 2026-01-26

Accept (Poster)